# Coordinated regulation of Rel expression by MAP3K4, CBP, and HDAC6 controls phenotypic switching

Noha Ahmed Mohammed Shendy [1,2], Deepthi Raghu[1], Sujoy Roy[3], Charles Hamilton Perry[1], Adiba Safi[1], Miguel Ramos Branco [4], Ramin Homayouni [3] & Amy Noel Abell [1✉]

Coordinated gene expression is required for phenotypic switching between epithelial and mesenchymal phenotypes during normal development and in disease states. Trophoblast stem (TS) cells undergo epithelial-mesenchymal transition (EMT) during implantation and placentation. Mechanisms coordinating gene expression during these processes are poorly understood. We have previously demonstrated that MAP3K4-regulated chromatin modifiers CBP and HDAC6 each regulate thousands of genes during EMT in TS cells. Here we show that CBP and HDAC6 coordinate expression of only 183 genes predicted to be critical regulators of phenotypic switching. The highest-ranking co-regulated gene is the NF-κB family member *Rel*. Although NF-κB is primarily regulated post-transcriptionally, CBP and HDAC6 control Rel transcript levels by binding *Rel* regulatory regions and controlling histone acetylation. REL re-expression in mesenchymal-like TS cells induces a mesenchymal-epithelial transition. Importantly, REL forms a feedback loop, blocking HDAC6 expression and nuclear localization. Together, our work defines a developmental program coordinating phenotypic switching.

[1] Department of Biological Sciences, University of Memphis, Memphis, TN 38152, USA. [2] Department of Chemistry, Faculty of Science, Mansoura University, Mansoura 35516, Egypt. [3] Department of Foundational Medical Studies, Oakland University William Beaumont School of Medicine, Rochester, MI 48309-4482, USA. [4] Centre for Genomics and Child Health, Blizard Institute, Queen Mary University of London, London E1 2AT, UK. ✉email: anabell@memphis.edu

Phenotypic switching plays a critical role during normal development, and pathways controlling this switching are reactivated in many disease states[1]. Epithelial to mesenchymal transition (EMT) is a biological program controlling cellular phenotype[1,2]. During EMT, cells lose epithelial features including cell–cell adhesion and epithelial markers, and gain the expression of EMT-inducing transcription factors (TFs) and mesenchymal markers. Cells undergoing EMT acquire motility and/or invasiveness. Importantly, this process is reversible through a mesenchymal to epithelial transition (MET), demonstrating the enormous plasticity of cells. However, mechanisms promoting the switch from mesenchymal to epithelial states are more poorly understood compared to those driving EMT[1].

We use trophoblast stem (TS) cells to define mechanisms controlling phenotypic states[3,4]. We isolated these multipotent TS cells from wild-type (TS$^{WT}$) blastocysts and from embryos created from intercrosses of mice heterozygous for a point mutation in MAP3K4 that inactivates its kinase activity[5]. Loss of MAP3K4 activity in TS cells (TS$^{KI4}$ cells) induces an intermediate EMT state in which cells display both epithelial and mesenchymal properties[3]. Importantly, TS$^{KI4}$ cells maintain stemness defined by self-renewal, multipotency, and chimera competence[3,5]. MAP3K4 induces histone lysine acetyltransferase CREBBP (CBP) activity and inhibits histone deacetylase6 (HDAC6) expression and activity[3,4]. TS$^{KI4}$ cells lacking MAP3K4 activity have reduced CBP activity and increased HDAC6 expression and activity[3,4]. Previously, we showed that CBP and HDAC6 each individually control thousands of genes in TS cells by controlling promoter acetylation of histone H2B at lysine 5 (H2BK5Ac)[3,4]. CBP acetylates promoters and promotes the epithelial state, whereas HDAC6 deacetylates promoters resulting in EMT[3,4]. We predicted that CBP and HDAC6 co-regulated genes are critical regulators of phenotypic switching. However, genes controlled by both CBP and HDAC6 are undefined.

Herein, we performed bioinformatics analyses of new RNA-seq data and published anti-H2BK5Ac chromatin immunoprecipitation (ChIP)-seq data to identify genes co-regulated by MAP3K4, CBP, and HDAC6. Of the thousands of genes independently regulated by either CBP or HDAC6, only 183 genes were co-regulated by both CBP and HDAC6 mediated regulation of histone acetylation. Within these co-regulated genes, 12% were DNA-binding proteins that could be drivers of the epithelial maintenance program. The highest-ranking DNA-binding protein was the NF-κB family member *Rel*. This family of TFs is constitutively expressed at transcript and protein levels in most cell types and is regulated post-transcriptionally by a protein repressor found in the cytoplasm[6]. In contrast, we show that Rel transcript levels are controlled by both CBP and HDAC6. In epithelial TS$^{WT}$ cells, Rel expression is promoted by CBP binding to both *Rel* promoter and predicted enhancer regions. In mesenchymal-like TS$^{KI4}$ cells, HDAC6 is bound to *Rel* regulatory regions, deacetylating histones H2BK5 and H3K27. Rel transcript expression is reduced in mesenchymal-like TS$^{KI4}$ cells deficient in MAP3K4 kinase activity, and re-expression of REL in TS$^{KI4}$ cells induces MET. Finally, we show that REL expression induces a switch to an epithelial phenotype by promoting the expression of 71% of the CBP/HDAC6 co-regulated DNA-binding proteins. Further, REL directly binds the *Hdac6* promoter, represses Hdac6 expression, and reduces nuclear HDAC6 localization. In summary, we define a developmental program that coordinately regulates the expression of Rel to drive switching between mesenchymal and epithelial states in TS cells.

## Results

### MAP3K4, CBP, HDAC6, and H2BK5Ac co-regulated genes.

We have previously demonstrated individual roles for MAP3K4

regulation of CBP or HDAC6 to control cellular phenotype[3,4]. To identify genes whose expression is coordinately regulated by MAP3K4, CBP, HDAC6, and H2BK5Ac, we performed bioinformatics analyses of new RNA-seq data and published H2BK5Ac ChIP-seq data. Using RNA-seq, we compared transcript levels in control wild-type (TS$^{WT}$) cells and TS cells expressing kinase-inactive MAP3K4 (TS$^{KI4}$ cells). To identify CBP dependent genes, we measured transcript levels in TS$^{WT}$ cells expressing lentiviral shRNA for Crebbp (TS$^{WTCBPsh}$), finding 4838 CBP-dependent genes (Fig. 1a). HDAC6 expression and activity are increased in TS$^{KI4}$ cells[4]. We defined 4184 HDAC6 dependent genes using RNA-seq of TS$^{KI4}$ cells expressing lentiviral shRNA for Hdac6 (TS$^{KI4H6sh}$). The intersection of three pairwise comparisons revealed 756 genes that were co-regulated by MAP3K4, CBP, and HDAC6 (Fig. 1a). Using our published anti-H2BK5Ac ChIP-seq data, we identified genes having reduced H2BK5Ac in TS$^{KI4}$ cells relative to TS$^{WT}$ cells[3]. Among the 756 genes co-regulated by MAP3K4, CBP, and HDAC6, only 183 genes had differential H2BK5Ac (Fig. 1a). Functional enrichment analyses of these 183 genes using Gene Ontology classifications revealed enrichment of genes in several categories including Actin Binding, Cytoskeletal Protein Binding, and Sequence-specific DNA Binding, with 22 (12%) of the 183 genes being TFs (Fig. 1b and Supplementary Data 1). One goal from these analyses was to identify genes that might act as broad regulators of cellular programs controlling phenotypic states. Because TFs regulate expression of many genes, we further analyzed TFs with an RPKM value > 0.5 in TS$^{WT}$ cells in the above experiments (Supplementary Fig. 1a). In addition, TFs were analyzed for the degree of change represented by the RPKM ratio for the following pairwise comparisons: TS$^{KI4}$/TS$^{WT}$, TS$^{WTCBPsh}$/TS$^{WT}$, and TS$^{KI4H6sh}$/TS$^{KI4}$. Similarly, H2BK5Ac of gene promoters was represented as a ratio of TS$^{KI4}$/TS$^{WT}$. To prioritize co-regulated TFs, TS$^{KI4}$/TS$^{WT}$ H2BK5Ac and RNA-seq ratios were ranked using the Wilcoxon rank sum test. *Rel* showed the most robust decrease in both transcript and H2BK5Ac in TS$^{KI4}$ cells relative to TS$^{WT}$ cells (Fig. 1c). Transcripts of TFs identified by RNA-seq were examined using qPCR, confirming that 80% of these genes have reduced expression in TS$^{KI4}$ cells relative to TS$^{WT}$ cells (Supplementary Fig. 1b, c). Together, these data identified a group of TFs that may play key roles in promoting the epithelial state.

### Selective loss of NF-κB member Rel in TS$^{KI4}$ cells in EMT.

As the highest-ranking co-regulated TF, we examined the expression of *Rel* in TS$^{WT}$ and TS$^{KI4}$ cells. *Rel* is a member of the NF-κB family of TFs consisting of five members that are constitutively expressed in most cell types, being regulated at the protein level by binding to a cytoplasmic repressor[6]. Thus, we were surprised that *Rel* was regulated at the transcript level in TS cells. qPCR showed that Rel was the only NF-κB family member with altered transcript expression in TS$^{KI4}$ cells (Fig. 2a). Western blotting of cytoplasmic and nuclear lysates showed reduction of REL in the cytoplasm and complete loss in the nucleus of TS$^{KI4}$ cells (Fig. 2b). Comparison of expression and localization of the NF-κB family in TS$^{WT}$ cells showed that RELA and RELB were localized primarily to the cytoplasm, suggesting they were inactive (Fig. 2b). REL, p50, and p52 family members were located in the cytoplasm and nucleus with nuclear localization suggesting active REL, p50, and p52 (Fig. 2b). TS$^{KI4}$ cells showed reduced protein and loss of nuclear localization of REL and p52 (Fig. 2b–d). Total NF-κB activity was reduced by >50% in TS$^{KI4}$ cells (Fig. 2e). Together, these data suggested that REL functions as the dominant NF-κB family member in epithelial TS cells.

Removal of FGF4, heparin, and mouse embryonic fibroblast conditioned media results in loss of stemness in TS cells and initiation of EMT with differentiation to a mixed population of

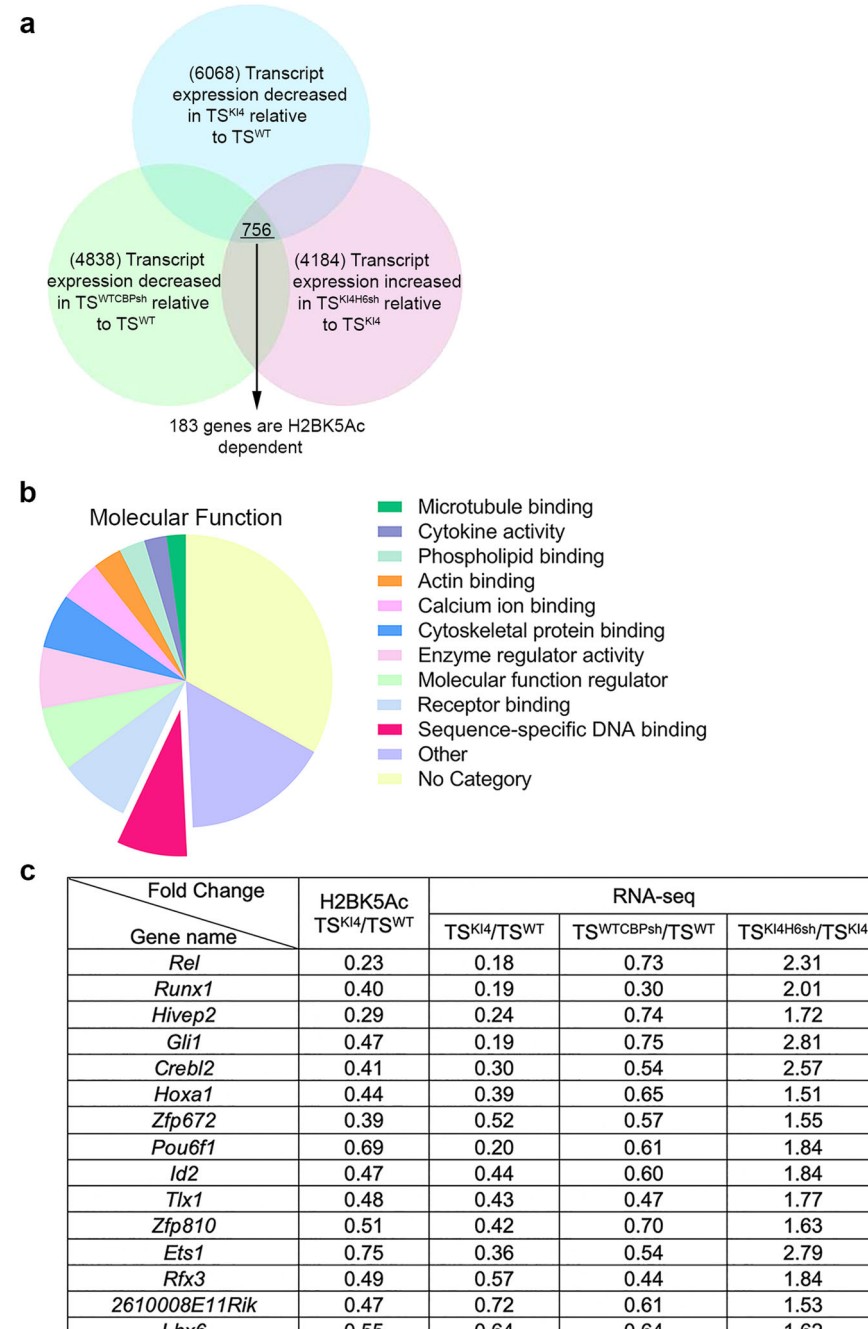

**Fig. 1 Analyses of genome wide RNA-seq and ChIP-seq data identify transcription factors whose expression is MAP3K4, CBP, HDAC6, and H2BK5Ac dependent.** **a** Bioinformatics analyses of RNA-seq and anti-H2BK5Ac ChIP-seq data from TS$^{WT}$ cells or TS$^{KI4}$ cells expressing control shRNA, TS$^{WT}$ cells expressing Crebbp shRNA (TS$^{WTCBPsh}$) or TS$^{KI4}$ cells expressing Hdac6 shRNA (TS$^{KI4H6sh}$). Venn diagram shows 756 genes whose expression is MAP3K4, CBP, and HDAC6 dependent. Of these 756 genes, 183 genes are H2BK5Ac dependent. The cutoff fold-change value for differentially expressed genes was ≤ 0.75 for TS$^{KI4}$/TS$^{WT}$ and TS$^{WTCBPsh}$/TS$^{WT}$ and ≥ 1.5 for TS$^{KI4H6sh}$/TS$^{KI4}$. **b** Enrichment of genes encoding DNA-binding proteins. Pie chart depicts the Gene Ontology for the molecular function annotation of the 183 co-regulated genes. Twelve percent of these genes are transcription factors (TFs) categorized under sequence-specific DNA binding. **c** Table shows differential expression of MAP3K4, CBP, HDAC6, and H2BK5Ac dependent TFs identified in (**b**). TFs with ≥ 0.5 reads per kilobase of transcript per million mapped reads (RPKM) in TS$^{WT}$ cells are shown. RNA-seq RPKM values for each sample were expressed as ratios. For example, RPKM values in TS$^{KI4}$ cells were divided by those for TS$^{WT}$ cells. Anti-H2BK5Ac ChIP-seq values determined using EpiCenter were also expressed as a ratio dividing values in TS$^{KI4}$ cells by those for TS$^{WT}$ cells[52]. TFs were prioritized using the Wilcoxon rank sum test of TS$^{KI4}$/TS$^{WT}$ H2BK5Ac ChIP-seq and RNA-seq ratios. See also Supplementary Fig. 1.

differentiated, mature trophoblasts (T$^{DIFF}$)[3]. One subtype of differentiated trophoblasts has acquired invasiveness (T$^{INV}$)[3]. As measured by qPCR, differentiation into mixed T$^{DIFF}$ resulted in only modest changes in NF-κB family member transcripts (Fig. 2f, g). In contrast, invasive T$^{INV}$ showed the selective induction of

RelA, RelB, and Nfκb2 transcripts, suggesting a role for these family members in invasive, differentiated trophoblasts (Fig. 2g). Importantly, Rel and Nfκb1 transcripts were not induced in invasive trophoblasts, consistent with an opposing role for REL to promote undifferentiated, epithelial TS cells (Fig. 2f).

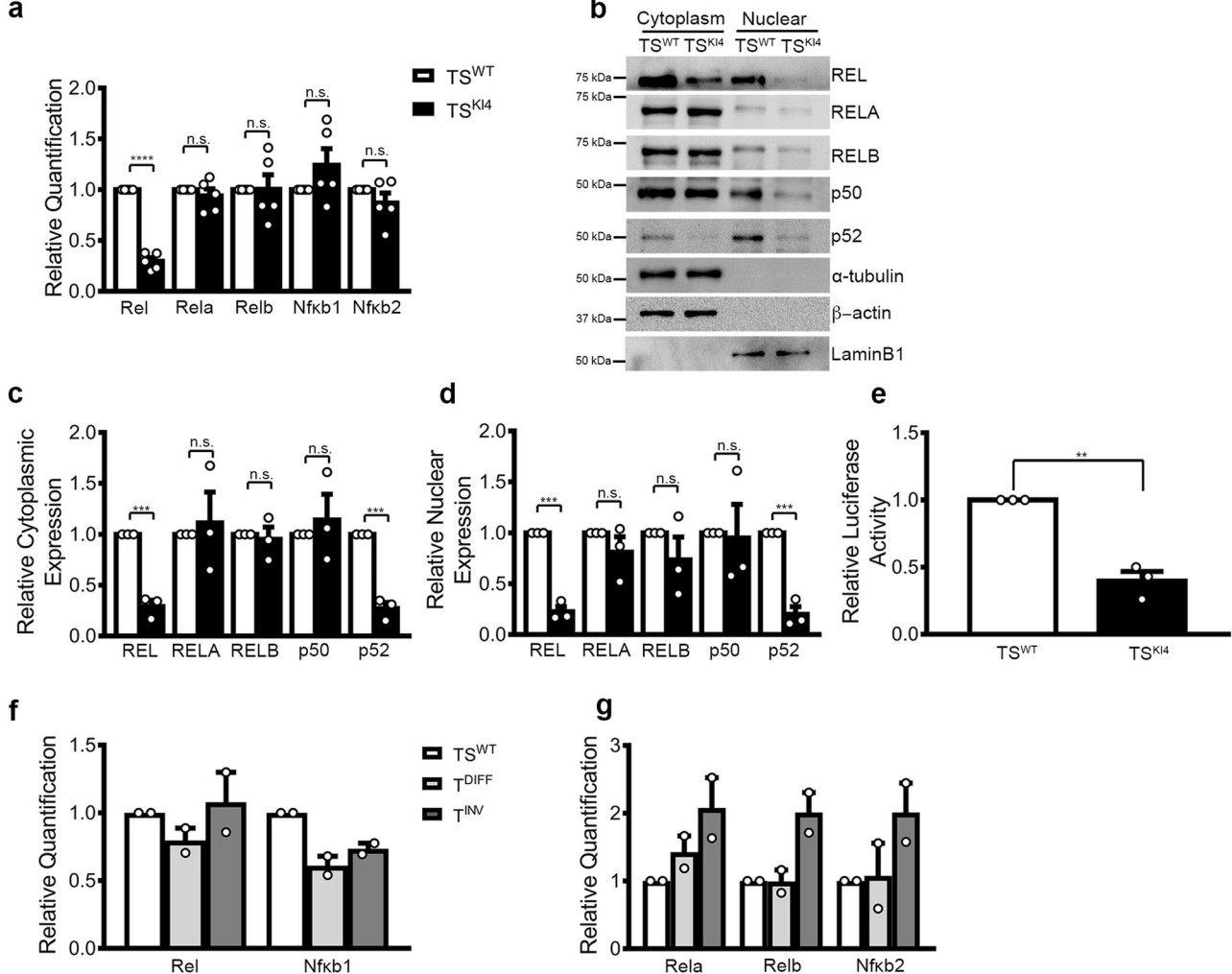

**Fig. 2 Selective regulation of REL expression in TS cells. a** Selective reduction of Rel transcript levels in TS^KI4 cells relative to TS^WT cells. Transcripts of all NF-κB members were measured by qPCR. The data normalized to Rps11 are expressed as a fold-change relative to TS^WT cells and are the mean ± SEM of n = 5 biologically independent experiments. **b** Cytoplasm and nuclear protein levels of NFκB members were measured using Western blotting in TS^WT cells and TS^KI4 cells. Blots are representative of n = 3 biologically independent experiments. **c, d** Densitometry was used to quantify cytoplasmic (**c**) and nuclear (**d**) NF-κB protein expression (**b**) from three biologically independent experiments. Data are expressed as fold-change relative to TS^WT cells and are the mean ± SEM of n = 3 biologically independent experiments. **e** NFκB transcriptional activity is reduced in TS^KI4 cells relative to TS^WT cells. Luciferase activity was measured in cells expressing a NF-κB luciferase construct. Data shown are the fold change in luciferase activity relative to TS^WT cells and are the mean ± SEM of n = 3 biologically independent experiments. **f, g** NF-κB member transcript levels in undifferentiated TS cells (TS^WT), TS cells differentiated for 4 days (T^DIFF), and invasive trophoblasts (T^INV) were measured using qPCR. The data normalized to Gapdh are expressed as a fold-change relative to undifferentiated TS^WT cells and are mean ± range of n = 2 biologically independent experiments. **p-value < 0.01; ***p-value < 0.001, ****p-value < 0.0001; Student's t test. (n.s. not significant). Western blots show cropped images. Uncropped images are available in Supplementary Materials.

**Induction of REL expression in TS^KI4 cells promotes MET.** To examine the role of REL in controlling the cellular phenotype, TS^KI4 cells were infected with a doxycycline-inducible construct expressing human *REL* and selected for stable transduction using puromycin. Addition of doxycycline (Dox) to cells transduced with human *REL* (TS^KI4R cells) resulted in modest expression of REL (Supplementary Fig. 2a, b). Methylation of lentiviral promoters can silence transgene expression, and inhibition of DNA methyltransferases can allow transgene expression[7,8]. Transient inhibition of DNA methyltransferases for 48 h with 5-Aza-2′-Deoxycytidine (Aza) in TS^KI4R cells did not affect expression of human REL transcript or protein (Supplementary Fig. 2a, b). However, the combined addition of Aza for 48 h and Dox for 96 h resulted in robust increases in both human REL transcript and protein measured using human REL specific primers and

antibody, respectively (Supplementary Fig. 2a, b). Further, transient co-treatment of TS^KI4R cells with Aza and Dox was sufficient to induce a phenotypic switch from mesenchymal-like to an epithelial-like morphology (Supplementary Fig. 2c). These data suggested that the induction of REL expression maybe sufficient to induce MET.

To study the effects of REL re-expression, we grew TS^KI4R cells continuously in the absence or presence of Dox. TS^KI4R cells grown in Dox showed robust expression of human REL transcript and protein (Fig. 3a, b). Importantly, TS^KI4R cells grown in Dox showed a stable, epithelial morphology compared to TS^KI4R cells cultured in the absence of Dox (Fig. 3c). TS^KI4R cells grown without Dox showed decreased expression of epithelial markers and increased expression of mesenchymal markers and EMT-inducing TFs relative to TS^WT cells (Fig. 3d–g). Culture of TS^KI4R

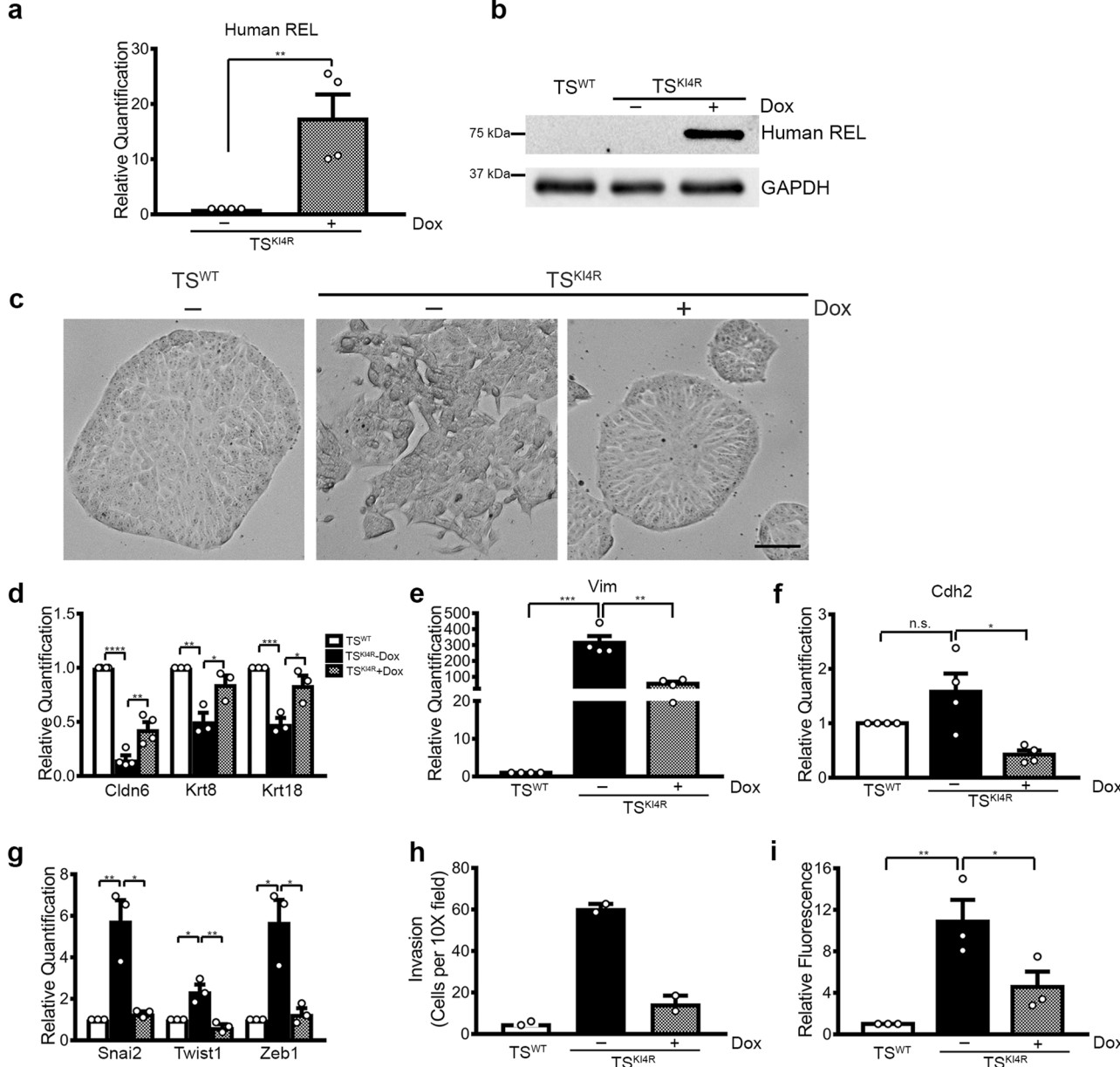

**Fig. 3 Induction of REL expression in mesenchymal-like TS^KI4 cells promotes an epithelial phenotype. a** Human REL transcript levels were measured in uninduced and Doxycycline (Dox) induced TS^KI4 cells transduced with lentiviral constructs expressing Tet-on-human *REL* (TS^KI4R) using qPCR and human specific REL primers. The data normalized to Rps11 are expressed as a fold-change relative to TS^KI4 cells and are the mean ± SEM of n = 4 biologically independent experiments. **b** REL protein expression was measured using Western blotting with anti-human specific REL antibody. Blots of whole-cell lysates are representative of n = 3 biologically independent experiments. **c** Dox induced REL expression restores an epithelial morphology in TS^KI4 cells re-expressing REL (TS^KI4R) cells. Representative phase microscopy images for n = 3 biologically independent experiments are shown. Scale bar represents 400 μm. **d** REL expression in TS^KI4 cells increased transcripts of epithelial markers. **e–g** Reduced mesenchymal markers (**e, f**) and EMT-inducing TFs (**g**) transcripts upon REL re-expression. **d–g** Transcripts were measured using qPCR. The data normalized to Rps11 are expressed as a fold-change relative to TS^WT cells and are the mean ± SEM. **d** n = 4 (Cldn6) and n = 3 (Krt8, Krt18) biologically independent experiments are shown. **e, f** n = 4 biologically independent experiments are shown. **g** n = 3 biologically independent experiments. **h** Invasion through growth factor reduced Matrigel was decreased in Dox induced TS^KI4R cells relative to uninduced TS^KI4R cells. Data show the mean ± range of n = 2 biologically independent experiments performed in triplicate. **i** Barrier formation is increased in TS^KI4 cells after REL re-expression. Data show the fold change in fluorescence of diffused dye relative to TS^WT cells and are the mean ± SEM of n = 3 biologically independent experiments. *p-value < 0.05; **p-value < 0.01; ***p-value < 0.001, ****p-value < 0.0001; Student's t test. (n.s. not significant). Western blots show cropped images. Uncropped images are available in Supplementary Materials. See also Supplementary Figs. 2 and 3.

cells continuously in Dox induced re-expression of epithelial markers (Fig. 3d, Supplementary Fig. 2d, e). Although expression of the mesenchymal marker Vim was only partially decreased by REL re-expression, expression of Cdh2 and EMT-inducing TFs Snai2, Twist1, and Zeb1 were returned to low levels observed in

TS^WT cells (Fig. 3e–g). Mesenchymal-like TS^KI4 cells show increased invasiveness and decreased barrier function relative to epithelial TS^WT cells (Fig. 3h, i). REL re-expression in Dox treated TS^KI4R cells repressed cellular invasion and promoted barrier function (Fig. 3h, i). Consistently, Dox treated TS^KI4R cells

showed reduced motility compared to untreated $TS^{KI4R}$ cells as measured by live cell imaging (Supplementary Movies 1–3). Together, these data indicated that REL re-expression induced a phenotypic change from mesenchymal to epithelial.

We also examined the impact of REL re-expression using a construct that constitutively expresses *REL*. Similar to Dox-induction, constitutive expression of REL in two different clones of $TS^{KI4}$ cells ($TS^{KI4R1}$ and $TS^{KI4R2}$) promoted an epithelial morphology (Supplementary Fig. 3a, b). $TS^{KI4R1}$ and $TS^{KI4R2}$ cells re-expressed epithelial markers and showed reduced expression of Vim and EMT-inducing TFs (Supplementary Fig. 3c–e). Constitutive REL expression also repressed invasion and partially restored barrier formation (Supplementary Fig. 3f, g). Microarray analyses of upregulated genes in $TS^{KI4R}$ cells relative to control infected $TS^{KI4}$ cells showed enrichment for genes in several categories important for cellular morphology including Focal adhesion and Cell–substrate adherens junction (Supplementary Fig. 3h). Together, these data suggested that REL re-expression in $TS^{KI4}$ cells induces MET.

**RELB expression in $TS^{KI4}$ cells induces a mesenchymal state**. We wondered if the effects of REL expression on $TS^{KI4R}$ cells were selective for REL, or if other NF-κB family members would have a similar effect. We infected $TS^{KI4}$ cells with lentivirus encoding human *RELB*, and selected for stable expression using puromycin ($TS^{KI4RB}$ cells). Human RELB expression was detected in $TS^{KI4RB}$ cells (Fig. 4a). RELB expression induced the transition of $TS^{KI4RB}$ cells to an entirely mesenchymal state (Fig. 4b). Consistent with changes in morphology, $TS^{KI4RB}$ cells showed the near complete loss of epithelial markers Cdh1 and Cldn6 (Fig. 4c). Further, Vim, Cdh2, and EMT-inducing TF expression were increased in $TS^{KI4RB}$ cells (Fig. 4d–f). In addition, cellular motility and invasiveness in $TS^{KI4RB}$ cells was greater than $TS^{KI4}$ cells, and barrier formation remained impaired (Fig. 4g, h, Supplementary Movie 4). These data showed that expression of RELB in $TS^{KI4}$ cells induced a complete EMT. Together, these data demonstrated the selectivity of NF-κB family members in control of TS cell phenotypic states.

**CBP directly binds promoter and enhancer regions of *Rel***. Based on our new RNA-seq data and published anti-H2BK5Ac ChIP-seq data, we identified *Rel* as a MAP3K4, CBP, HDAC6, and H2BK5Ac co-regulated gene. To determine if the effects of CBP on Rel expression are through direct binding to and control of the *Rel* gene, we examined Rel expression in $TS^{WT}$ cells and $TS^{WT}$ cells expressing shRNAs for Crebbp ($TS^{WTCBPsh1}$ and $TS^{WTCBPsh2}$). Lentiviral shRNAs for Crebbp resulted in an 82% reduction in Crebbp transcript (Fig. 5a). CBP knockdown in $TS^{WTCBPsh1}$ and $TS^{WTCBPsh2}$ cells reduced Rel transcript and protein relative to $TS^{WT}$ cells (Fig. 5b–d). *Rel* was the only NF-κB family member whose expression was decreased with CBP knockdown (Fig. 5b–d). To determine if CBP bound the *Rel* promoter, we performed ChIP with anti-CBP antibody. We measured a 15-fold enrichment of CBP on the *Rel* promoter (−700) compared to isotype control (IgG) antibody (Fig. 5e). Enrichment of CBP on the *Rel* promoter was decreased in $TS^{KI4}$ cells relative to $TS^{WT}$ cells (Fig. 5f). These data suggested that CBP is bound to the *Rel* promoter and loss of MAP3K4 kinase activity results in decreased binding of CBP to the *Rel* promoter.

CBP binds enhancer regions of many genes with CBP promoting gene expression[9]. We wondered if CBP was also bound to enhancer regions for *Rel*. To identify putative enhancer regions, we examined contact maps from published promoter capture Hi-C data from wild-type TS cells[10]. We chose fourteen sites based on number of reads and distance from the *Rel*

transcription start site (Supplementary Fig. 4a, b). Four of these sites were found in predicted promoters (Supplementary Fig. 4a, b). For the other sites, two were located upstream of the *Rel* promoter and eight were found downstream (Supplementary Fig. 4a, b). H3K27Ac marks promoters and active enhancers[11]. Comparison of IgG ChIP to anti-H3K27Ac ChIP at several of these sites revealed a 9 to 40-fold increase in enrichment in H3K27Ac at these sites (Supplementary Fig. 4c). Using anti-H3K27Ac ChIP-PCR, we compared H3K27Ac at all regions in $TS^{KI4}$ cells relative to $TS^{WT}$ cells. H3K27Ac did not change between $TS^{KI4}$ cells and $TS^{WT}$ cells for most of the contact sites suggested by promoter capture Hi-C data (Supplementary Fig. 4d, e). For promoters predicted to contact the *Rel* promoter, we observed only very modest decreases in H3K27Ac in $TS^{KI4}$ cells relative to $TS^{WT}$ cells (Supplementary Fig. 4d). The only exception was the predicted promoter for Gm12061-001. In contrast, there were dramatic changes in H3K27Ac at several potential enhancer sites predicted using promoter capture Hi-C. Surprisingly, the site −37106 bp upstream of the *Rel* promoter showed a two-fold increase in H3K27Ac in $TS^{KI4}$ cells relative to $TS^{WT}$ cells (Supplementary Fig. 4e). In contrast, a region found between +29050 to +35817 bp downstream of the *Rel* promoter showed approximately a 50% decrease in H3K27Ac in $TS^{KI4}$ cells relative to $TS^{WT}$ cells (Supplementary Fig. 4e). H3K27Ac of an additional region located between +83091 and +101930 downstream of the *Rel* promoter was also decreased in $TS^{KI4}$ cells (Supplementary Fig. 4e). Based on diminished H3K27Ac in $TS^{KI4}$ cells relative to $TS^{WT}$ cells, proximity to the *Rel* promoter, and contacts with the *Rel* promoter as measured by promoter capture Hi-C, we predicted that the region between +29050 to +35817 bp may serve to enhance *Rel* expression. We measured a 9–15-fold enrichment of CBP at these sites relative to IgG antibody, suggesting CBP was bound to these sites in $TS^{WT}$ cells (Fig. 5e). However, anti-CBP ChIP-PCR comparing CBP enrichment in $TS^{KI4}$ cells relative to $TS^{WT}$ cells did not show differences in CBP binding at these sites (Fig. 5f). Together, these data suggested that CBP is bound to these putative *Rel* enhancer sites, but altered binding of CBP was not responsible for changes in H3K27Ac at these sites in $TS^{KI4}$ cells relative to $TS^{WT}$ cells.

**HDAC6 represses histone acetylation on *Rel* regulatory sites**. HDAC6 expression and activity are increased in $TS^{KI4}$ cells relative to $TS^{WT}$ cells[4]. As shown in Fig. 6a, Hdac6 transcripts were increased in $TS^{KI4}$ cells. shRNA knockdown of Hdac6 in $TS^{KI4H6sh}$ cells reduced Hdac6 transcripts to levels lower than $TS^{WT}$ cells (Fig. 6a). $TS^{KI4H6sh}$ cells had increased Rel transcripts and protein (Fig. 6b, c). Treatment of $TS^{KI4}$ cells with the HDAC6 selective inhibitor Tubastatin also increased Rel transcripts (Fig. 6d). Together, these data suggested that HDAC6 represses Rel expression.

We have previously shown that HDAC6 in the nuclei of $TS^{KI4}$ cells binds and represses the promoters of genes encoding tight junction proteins like Claudin6[4]. However, the expression of these genes is CBP independent (Supplementary Table 1). To determine if HDAC6 directly regulates *Rel* expression, we performed anti-HDAC6 ChIP-PCR. HDAC6 was enriched on the *Rel* promoter and predicted enhancer regions in $TS^{KI4}$ cells, and Hdac6 knockdown reduced enrichment (Fig. 6e, Supplementary Fig. 4f). These data suggested that HDAC6 is physically bound to the regulatory regions of *Rel*. Total H2BK5Ac was reduced in $TS^{KI4}$ cells, and shRNA knockdown of Hdac6 restored H2BK5Ac (Fig. 6f). Anti-H2BK5Ac ChIP-PCR showed reduced H2BK5Ac at the promoter and predicted enhancer regions of *Rel* in $TS^{KI4}$ cells (Fig. 6g). Acetylation of these sites was HDAC6 dependent, because knockdown of Hdac6 restored H2BK5Ac to

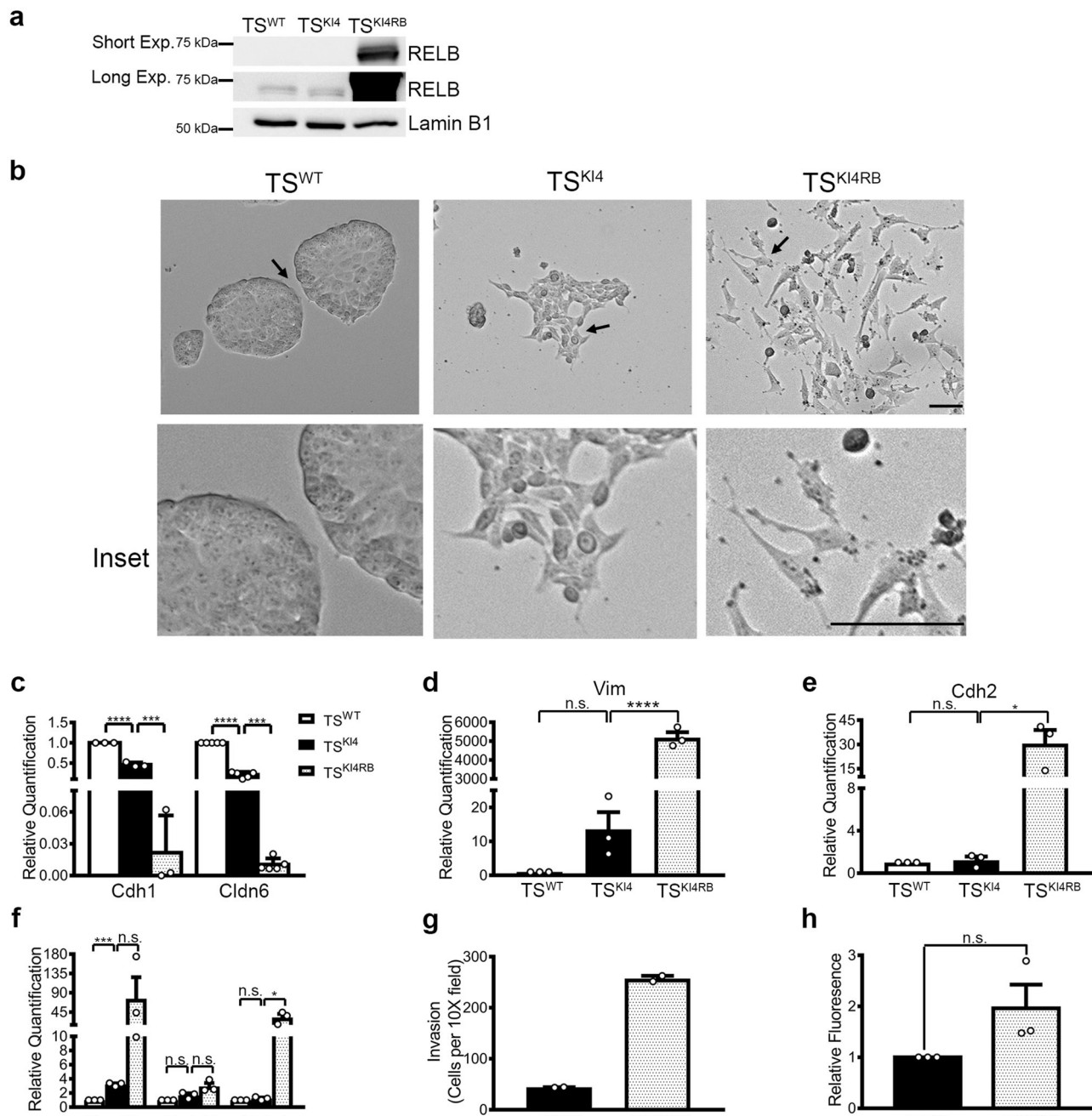

**Fig. 4 Expression of RELB in TS<sup>KI4</sup> cells induces EMT. a** RELB protein levels were measured in TS<sup>WT</sup> cells, TS<sup>KI4</sup> cells, and TS<sup>KI4</sup> cells expressing RELB (TS<sup>KI4RB</sup>) cells. Western blots of nuclear lysates are representative of $n = 2$ biologically independent experiments. **b** RELB expression induces a mesenchymal morphology in TS<sup>KI4R</sup> cells. Representative phase microscopy images for $n = 3$ biologically independent experiments are shown. Scale bar represents 400 μm. Arrows show the area of enlarged insets. **c** RELB expression in TS<sup>KI4</sup> cells results in loss of epithelial marker expression. Transcripts were measured using qPCR. The data normalized to Rps11 are expressed as fold-change relative to TS<sup>WT</sup> cells and are the mean ± SEM of $n = 3$ (Cdh1) or $n = 5$ (Cldn6) biologically independent experiments. **d–f** Increased transcripts of mesenchymal markers (**d**, **e**) and EMT-inducing TFs (**f**) upon RELB expression. Transcripts were measured using qPCR. The data normalized to Rps11 are expressed as fold-change relative to TS<sup>WT</sup> cells and are the mean ± SEM of $n = 3$ biologically independent experiments. **g** Invasion through growth factor reduced Matrigel was increased in TS<sup>KI4RB</sup> cells relative to TS<sup>KI4</sup> cells. Data show the mean ± range of $n = 2$ biologically independent experiments performed in triplicate. **h** Barrier formation shown as the fold change in fluorescence of diffused dye relative to TS<sup>KI4</sup> cells. Data are expressed as the mean ± SEM of $n = 3$ biologically independent experiments. *$p$-value < 0.05; ***$p$-value < 0.001, ****$p$-value < 0.0001; Student's $t$ test. (n.s. not significant). Western blots show cropped images. Uncropped images are available in Supplementary Materials.

wild-type levels (Fig. 6g). Based on our new findings of changes in H3K27Ac on predicted enhancer regions in TS<sup>KI4</sup> cells relative to TS<sup>WT</sup> cells, we examined total H3K27Ac, finding drastically diminished total H3K27Ac in TS<sup>KI4</sup> cells that was restored to

wild-type levels by knockdown of Hdac6 (Fig. 6f). Consistently, H3K27Ac on the *Rel* promoter was reduced and knockdown of *Hdac6* restored H3K27Ac (Fig. 6h). However, loss of Hdac6 had only modest effects on H3K27Ac at predicted *Rel* enhancer

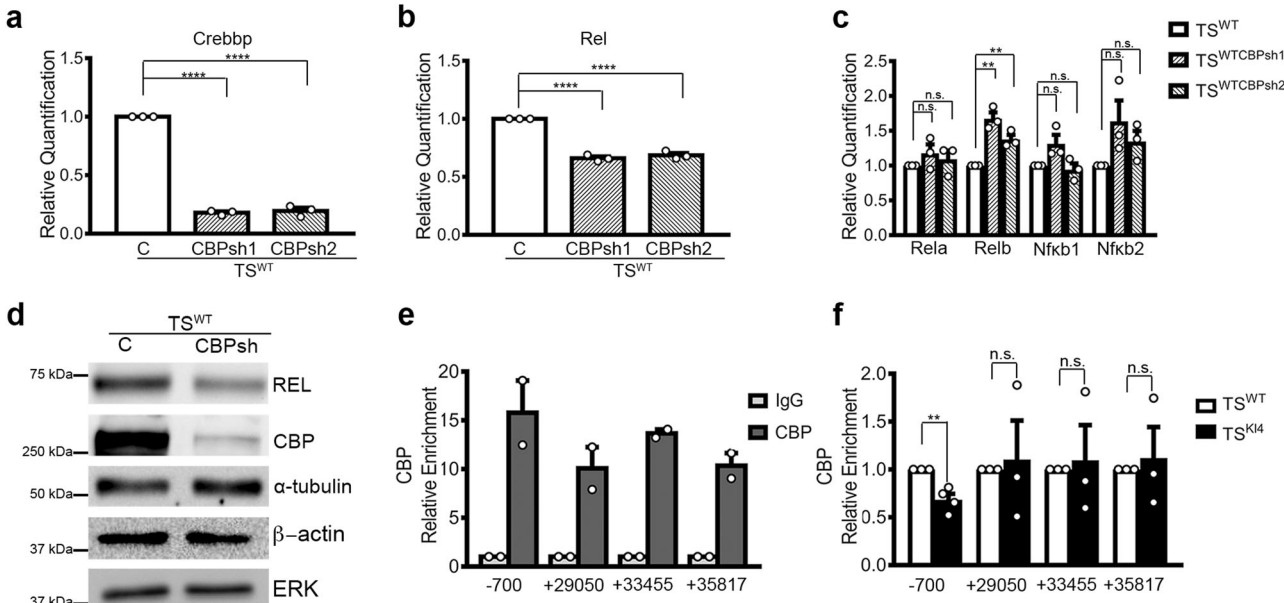

**Fig. 5 CBP promotes REL expression in TS cells. a** Crebbp transcripts levels were measured in TS$^{WT}$ cells and TS$^{WT}$ cells expressing two independent Crebbp shRNAs (TS$^{WTCBPsh1}$ and TS$^{WTCBPsh2}$) using qPCR. **b** shRNA knockdown of CBP results in decreased Rel transcripts measured using qPCR. **c** Transcript levels of the other NF-κB members were measured using qPCR. **a–c** qPCR data normalized to Rps11 are expressed as a fold-change relative to TS$^{WT}$ cells and are the mean ± SEM of $n = 3$ biologically independent experiments. **d** CBP knockdown results in reduced REL protein expression. Western blots are representative of $n = 3$ biologically independent experiments. **e** CBP enrichment on the Rel promoter (−700) and predicted enhancer regions (+29050, +33455, and +35817) was measured using anti-CBP ChIP-PCR. Data are expressed as a fold change in enrichment of anti-CBP ChIP relative to anti-IgG ChIP. Data represent the mean ± range of $n = 2$ biologically independent experiments. **f** Anti-CBP ChIP-PCR shows decreased CBP enrichment on the Rel promoter in TS$^{KI4}$ cells relative to TS$^{WT}$ cells. Experiments were performed as in (**e**). Data show the mean ± SEM of $n = 4$ (promoter) or $n = 3$ (enhancers) biologically independent experiments. **p-value < 0.01; ****p-value < 0.0001; Student's t test. (n.s. not significant). Western blots show cropped images. Uncropped images are available in Supplementary Materials. See also Supplementary Fig. 4.

regions (Fig. 6h). Together, these data suggest that HDAC6 represses Rel expression by promoting the deacetylation of both H2BK5 and H3K27 on Rel regulatory regions.

As human HDAC6 contains a tetradecapeptide repeat domain not present in mouse HDAC6 that promotes cytoplasmic retention of human HDAC6, we wondered if HDAC6 regulates REL expression in human cells[12]. Comparison of human mammary epithelial cells (HMECs) to mesenchymal claudin-low SUM159 breast cancer cells revealed decreases in REL transcript and increases in RELB transcript in SUM159s relative to HMECs (Supplementary Fig. 5a–c). Nuclear HDAC6 and RELB protein levels were also increased in SUM159s (Supplementary Fig. 5d). ChIP-PCR showed enrichment of HDAC6 on the REL promoter and a concomitant decrease in H2BK5Ac on the REL promoter in SUM159s (Supplementary Fig. 5e, f). These data suggest that HDAC6 represses REL expression in both mouse and human cells.

**REL promotes the re-expression of co-regulated TFs**. A key remaining question was the mechanism by which REL re-expression in mesenchymal-like TS$^{KI4}$ cells induced MET. Further, how does RELB expression induce EMT? Previous studies have demonstrated the critical importance of NF-κB induction of inflammatory proteins to promote EMT[13,14]. Thus, one possible explanation for the outcomes of REL versus RELB expression on the cellular phenotype of TS$^{KI4}$ cells maybe due to differential expression of inflammatory proteins. RNA-seq data showed the expression of very few genes encoding inflammatory cytokines or their receptors (Supplementary Data 2). However, we noted modest changes in the transcripts for Tnf and Tnfsf12 (TWEAK) and their receptors in TS$^{KI4}$ cells relative to TS$^{WT}$ cells (Supplementary Data 2). As measured by qPCR, REL expression did

not induce the expression of Tnf or Tnfsf12 (Supplementary Fig. 6a, b). In contrast, RELB expression induced robust increases in Tnfsf12 and its receptor Tnfrsf12a, suggesting that EMT induction by RELB expression may involve expression of inflammatory cytokines and receptors (Supplementary Fig. 6a, b).

Since REL re-expression in TS$^{KI4}$ cells did not induce inflammatory proteins, we wondered if REL re-expression induced MET by influencing the expression of the other fourteen TFs co-regulated by MAP3K4, CBP, and HDAC6 (Fig. 1c). Examination of microarray data from TS$^{KI4R}$ cells constitutively re-expressing REL revealed that 71% (10/14) of the co-regulated TFs were also re-expressed in TS$^{KI4R}$ cells relative to control TS$^{KI4}$ cells (Fig. 7a). However, expression of only three TFs, Ets1, Hivep2, and Id2, was statistically significant (Fig. 7a). To further examine the impact of REL re-expression on the expression of these other TFs, we performed qPCR. Although we were unable to detect Lhx6 using multiple primer pairs, we found increased expression of ten of the fourteen co-regulated TFs in two individual TS$^{KI4}$ cell clones re-expressing REL (TS$^{KI4R1}$ and TS$^{KI4R2}$ cells) (Fig. 7b–d). Although some TFs were only modestly changed, expression of many of the TFs was restored to TS$^{WT}$ cell levels (Fig. 7b–d). Further, transcripts for Ets1 and Zfp672 were 3–4 fold higher in TS$^{KI4R}$ cells than TS$^{WT}$ cells (Fig. 7b). These data suggested that REL controlled the expression of most of the MAP3K4, CBP, HDAC6, and H2BK5Ac co-regulated TFs.

**REL controls Hdac6 expression and localization promoting MET**. Because REL promoted the re-expression of most of the co-regulated TFs, we wondered if REL controlled the expression of the histone modifiers CBP and/or HDAC6. As measured by qPCR, REL re-expression did not induce Crebbp expression

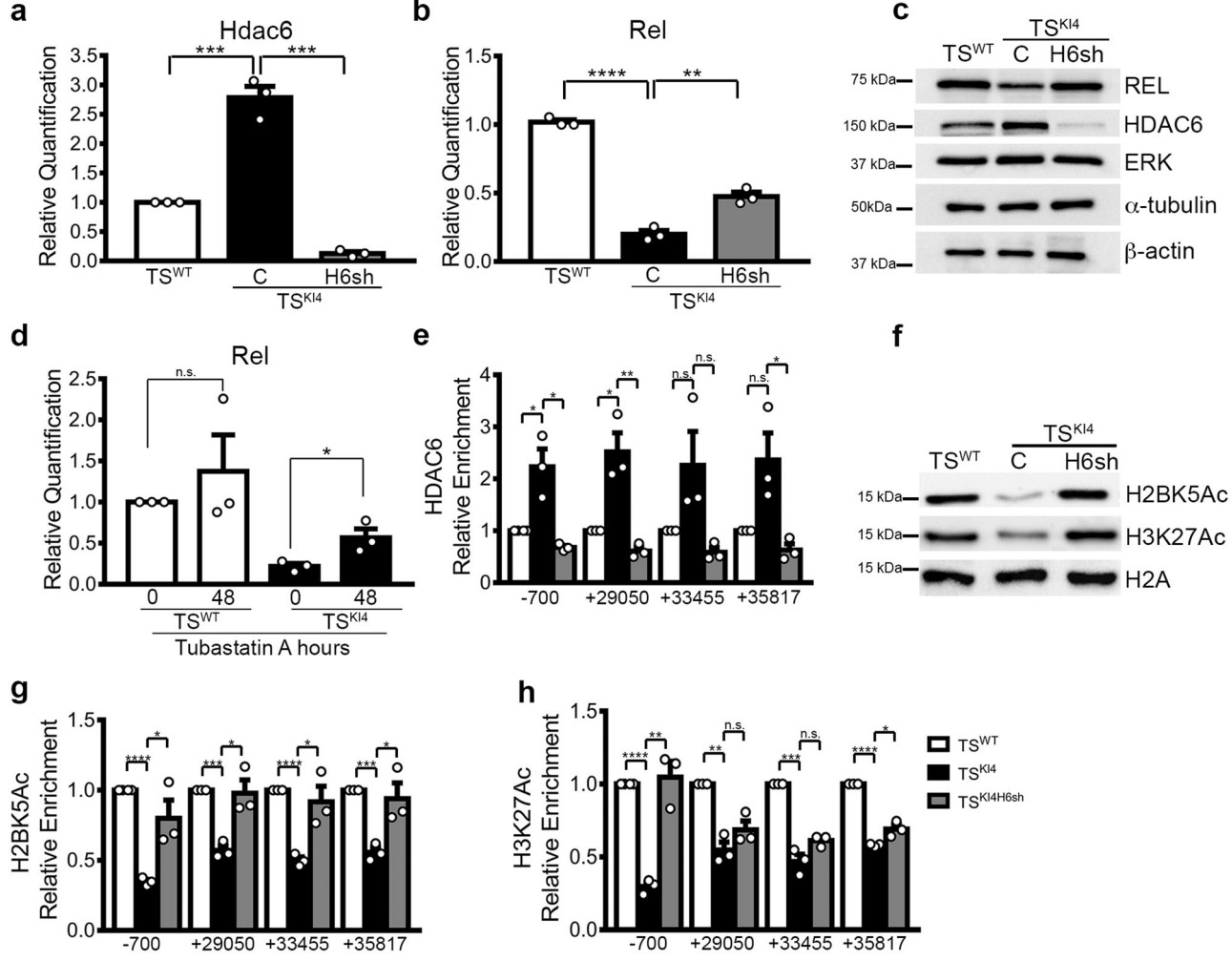

**Fig. 6 HDAC6 regulates Rel expression by controlling H2BK5Ac and H3K27Ac on the *Rel* promoter and predicted enhancer regions. a** Hdac6 transcript levels in TS$^{WT}$ cells and TS$^{KI4}$ cells expressing control shRNA (C) or TS$^{KI4}$ cells expressing Hdac6 shRNA (TS$^{KI4H6sh}$) were measured using qPCR. **b** HDAC6 knockdown in TS$^{KI4H6sh}$ cells increases Rel transcript levels relative to TS$^{KI4}$ cells. **c** HDAC6 knockdown increases REL protein expression. Western blots of whole-cell lysates are representative of $n = 3$ biologically independent experiments. **d** Inhibition of HDAC6 activity using Tubastatin A partially restores Rel expression at transcript level in TS$^{KI4}$ cells. Cells were treated for 48 h with DMSO (0) or 10 μM Tubastatin A. **a**, **b**, **d** qPCR data normalized to Rps11 are the mean ± SEM of $n = 3$ biologically independent experiments. **e** Anti-HDAC6 ChIP-PCR shows increased HDAC6 enrichment on the *Rel* promoter (−700) and predicted enhancer regions (+29050, +33455, and +35817) in TS$^{KI4}$ cells relative to TS$^{WT}$ cells. Data shown are the mean ± SEM of n = 3 biologically independent experiments. **f** HDAC6 knockdown restores total H2BK5Ac and H3K27Ac in TS$^{KI4}$ cells. Western blots are representative of $n = 3$ biologically independent experiments. **g** H2BK5Ac of the *Rel* promoter and predicted enhancer regions is completely restored upon HDAC6 knockdown in TS$^{KI4}$ cells as measured by anti-H2BK5Ac ChIP-PCR. **h** H3K27Ac is completely restored on the *Rel* promoter as measured by anti-H3K27Ac ChIP-PCR. **g**, **h** Data shown are the mean ± SEM of $n = 3$ biologically independent experiments. *$p$-value < 0.05; **$p$-value < 0.01; ***$p$-value < 0.001, ****$p$-value < 0.0001; Student's $t$ test. (n.s. not significant). Western blots show cropped images. Uncropped images are available in Supplementary Materials. See also Supplementary Figs. 4 and 5.

(Fig. 7e). However, both transient and stable inducible and constitutive REL expression decreased Hdac6 transcripts (Fig. 7f, Supplementary Fig. 6c, d). Although HDAC6 protein expression remained elevated in the cytoplasm of TS$^{KI4R}$ cells, nuclear localization of HDAC6 was dramatically reduced in Dox induced TS$^{KI4R}$ cells relative to untreated TS$^{KI4R}$ cells (Fig. 7g, Supplementary Fig. 6e). Western blotting of whole-cell lysates showed decreased total HDAC6 protein in cells re-expressing REL (Fig. 7h). Together, these data indicated that REL expression altered HDAC6 expression and localization. To determine if REL directly controls HDAC6 expression, we performed anti-REL ChIP-PCR. Although REL was not enriched on the *Hdac6* promoter in TS$^{KI4}$ cells, REL was enriched six-fold on the *Hdac6* promoter in TS$^{WT}$ cells relative to IgG (Fig. 7i). Together, these data suggest that REL expression induces a phenotypic switch

from mesenchymal to epithelial in part by repressing HDAC6 expression.

## Discussion

Herein, we define a program controlling phenotypic switching in stem cells from mesenchymal to epithelial states. Although each individually regulates thousands of genes, together MAP3K4, CBP, and HDAC6 coordinate promoter H2BK5Ac and the expression of a specific set of 183 genes to regulate phenotype. We demonstrate the previously unknown role of the co-regulated gene, *Rel*, in inducing the epithelial state. Mesenchymal-like TS cells deficient in MAP3K4 kinase activity have reduced Rel expression, and re-expression of REL induces a mesenchymal to epithelial switch. This effect is selective for REL, as expression of

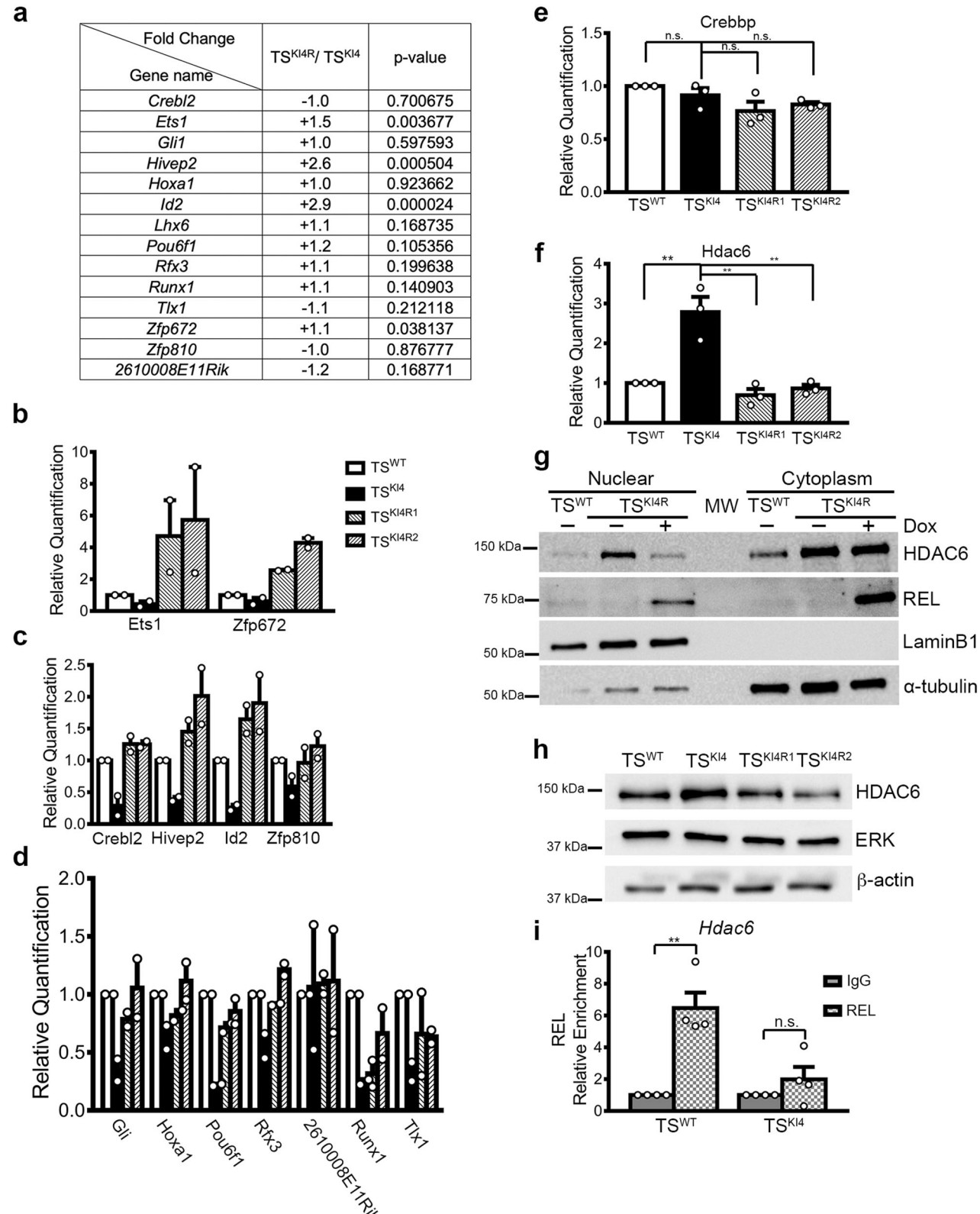

RELB in TS^KI4 cells induces a fully mesenchymal state. CBP and HDAC6 bind directly to the regulatory regions of *Rel*, controlling the acetylation of histones H2BK5 and H3K27. In addition to the epigenetic regulation of *Rel* by CBP and HDAC6, we show a reciprocal regulation of HDAC6 where REL functions to repress HDAC6. Together, our work defines a developmental program

that coordinates gene expression for controlling phenotypic switching between mesenchymal and epithelial states (Fig. 8).

We have discovered previously unidentified roles for NF-κB family members in TS cells isolated from mouse pre-implantation blastocysts. We have found that Rel expression induces MET, whereas expression of Relb induces EMT, demonstrating selective

**Fig. 7 REL re-expression induces MET by decreasing nuclear HDAC6 levels and increasing the expression of MAP3K4, CBP, and HDAC6 co-regulated transcription factors. a** REL re-expression in TS[KI4R] cells promotes the re-expression of co-regulated TFs. Microarray data from TS[KI4] cells and TS[KI4] expressing human *REL* under a constitutive promoter (TS[KI4R] cells) were examined for expression changes in co-regulated TFs. Data are expressed as a log2 fold change in TS[KI4R] cells relative to TS[KI4] cells. Statistical significance of fold changes expressed as p-values was determined using a one-way ANOVA. **b–d** Increased transcript levels for most of the co-regulated TFs upon re-expression of REL. Control infected TS[WT] cells and TS[KI4] cells, or two independent clones of TS[KI4] cells constitutively expressing REL TS[KI4R1] and TS[KI4R2] cells were examined. qPCR data show the mean ± range of $n = 2$ biologically independent experiments performed in triplicate. **e** Crebbp expression in TS[KI4] cells constitutively expressing REL. qPCR show the mean ± SEM of $n = 3$ biologically independent experiments. **f** Hdac6 transcript expression in TS[KI4] cells constitutively expressing REL. qPCR data show the mean ± SEM of $n = 3$ biologically independent experiments. **g** Decreased HDAC6 protein in the nuclei of TS[KI4R] cells re-expressing REL. Protein expression was measured in cytoplasmic and nuclear extracts. Data show representative images from $n = 3$ biologically independent experiments. **h** Decreased total HDAC6 protein in TS[KI4] cells constitutively expressing REL measured using Western blotting of whole-cell lysates. Data show representative images from $n = 2$ biologically independent experiments. **i** Anti-REL ChIP-PCR shows increased enrichment of REL relative to isotype control IgG on the *Hdac6* promoter in TS[WT] cells. Data shown are the mean ± SEM of $n = 4$ biologically independent experiments. *p-value < 0.05; **p-value < 0.01; ***p-value < 0.001, ****p-value < 0.0001; Student's *t* test. (n.s. not significant). Western blots show cropped images. Uncropped images are available in Supplementary Materials. See also Supplementary Fig. 6.

roles for NF-κB family members in TS cells. Previous work has shown the presence of Rel, Rela, Relb, and NFκB1 transcripts in mouse oocytes, and two-cell, eight-cell, and blastocyst stage embryos[15]. In cultured mouse embryonic stem (ES) cells isolated from blastocysts, NF-κB activity is inhibited and differentiation of ES cells induces NF-κB activity[16]. Mouse ES cells express very low levels of REL, and overexpression of REL in mouse ES cells induces differentiation and a dramatic loss of the epithelial state[17]. Similar to ES cells, we show that TS cells express all five NF-κB family members. However, based on nuclear localization, REL is the primary active family member in TS cells. Differentiation of TS cells to invasive trophoblasts specifically induces the expression of Rela, Relb, and NF-κB2, suggesting selective roles for specific NF-κB family members in invasive trophoblasts. Unlike ES cells where REL overexpression induces differentiation, re-expression of REL in mesenchymal-like cells promotes an epithelial phenotype, suggesting a different role for REL in TS cells. Our examination of published anti-CDX2 and anti-EOMES ChIP-seq data revealed binding of both stemness promoting TFs to the *Rel* promoter in wild-type TS cells[18–21]. These findings are consistent with Rel expression in TS cells and a role for REL in promoting the epithelial phenotype. Evaluation of ChIP-seq data from Rugg-Gunn et al. showed the presence of repressive H3K27me3 marks on the *Rel* promoter in mouse ES cells that is absent in TS cells[22]. In contrast, the *Rel* promoter is enriched for active H3K4me3 marks in TS cells[22]. These findings are consistent with the lack of Rel expression in ES cells, and the presence of Rel expression in TS cells. Together, these results indicate very different roles for REL in embryonic stem cells versus extra-embryonic TS cells.

Although we and others have defined the roles of NF-κB family members in cells from blastocysts, it has been difficult to define their roles during development. The use of simpler model organisms including *Drosophila*, *Xenopus laevis*, and *Danio rerio* has enabled identification of the critical role of NF-κB in developmental programs like embryonic dorsal-ventral patterning[23]. Success of studies in *Drosophila* is due in part to having fewer NF-κB family members[24]. However, it has been extremely difficult to establish the precise roles of individual NF-κB family members during development in higher organisms. One limitation has been the relatively low numbers of phenotypes that occur with single deletion of NF-κB family members in mice[23,25]. The ability of family members to bind similar κB sites creates the potential for functional redundancy and compensation[6,26]. Signaling complexity is created by their ability to function as homodimers and heterodimers. Further, the family members often have opposing roles, and deletion of one member may induce a phenotype created by activity of another member[23,27,28]. Although single

deletion of *RelA*, *Ikkβ*, or *NEMO* results in embryonic lethality, deletion of two family members leads to more deleterious phenotypes with earlier onset. For example, the single deletion of *RelA* in mice results in embryonic lethality at E15 due to liver degeneration[29,30]. Deletion of *Rel* does not result in embryonic lethality[31,32]. However, co-deletion of *RelA* and *Rel* results in earlier lethality at E13.5, suggesting REL may compensate for RELA[25]. Analysis of more complex genetic alterations of NF-κB family members in mice has established roles for NF-κB family members in neural tube closure, and in the development of the lung, muscle, skin, skeleton, and hematopoietic system[23]. Using TS[KI4] cells isolated from mice with the targeted inactivation of MAP3K4, we have dissected the specific role of REL in TS cells. Although all members are expressed, NF-κB family members cannot compensate for loss of REL in undifferentiated TS cells. This lack of compensation may be due to the role of RELA and RELB in differentiated, invasive trophoblasts.

Our new work demonstrates that Rel transcript levels are epigenetically regulated by both CBP and HDAC6 mediated promoter and enhancer acetylation during phenotypic switching in TS cells. This finding contrasts strongly with regulation of NF-κB family members primarily at the level of activity by controlling the translocation of constitutively expressed NF-κB dimers between the cytoplasm and nucleus[6]. In contrast, we show that CBP binds to *Rel* regulatory regions and promotes *Rel* expression. Importantly, loss of MAP3K4 activity in mesenchymal-like TS[KI4] cells reduces binding of CBP to the *Rel* promoter and decreases the expression of *Rel*. Further, we show that HDAC6 represses *Rel* expression by direct binding to *Rel* regulatory regions and decreasing H2BK5 and H3K27 acetylation on both *Rel* promoter and predicted enhancer regions. Roles for HDAC6 in deacetylation of H2BK5 on enhancers has not been previously demonstrated. Further, we identify a previously unknown role for HDAC6 in deacetylation of H3K27 on both promoters and distal regulatory regions. Together, our work defines a mechanism for epigenetic co-regulation of *Rel* expression by CBP and HDAC6, controlling phenotypic switching.

NF-κB family members have been shown to induce EMT in several systems, especially in cancer[33,34]. Induction of EMT by NF-κB is often dependent on expression of inflammatory cytokines and their receptors[13,14,35]. In this inflammatory context, NF-κB promotes the expression of several EMT-inducing TFs, including Snai1, Snai2, Twist1, Zeb1, and Zeb2[28,35–39]. Similarly, expression of RELB in TS[KI4] cells induces the expression of the inflammatory genes Tnfsf12 and Tnfrsf12a and EMT. In contrast, we show for the first time to our knowledge that REL expression induces MET. Importantly, REL expression in TS[KI4] cells does not induce expression of inflammatory proteins or EMT. Instead,

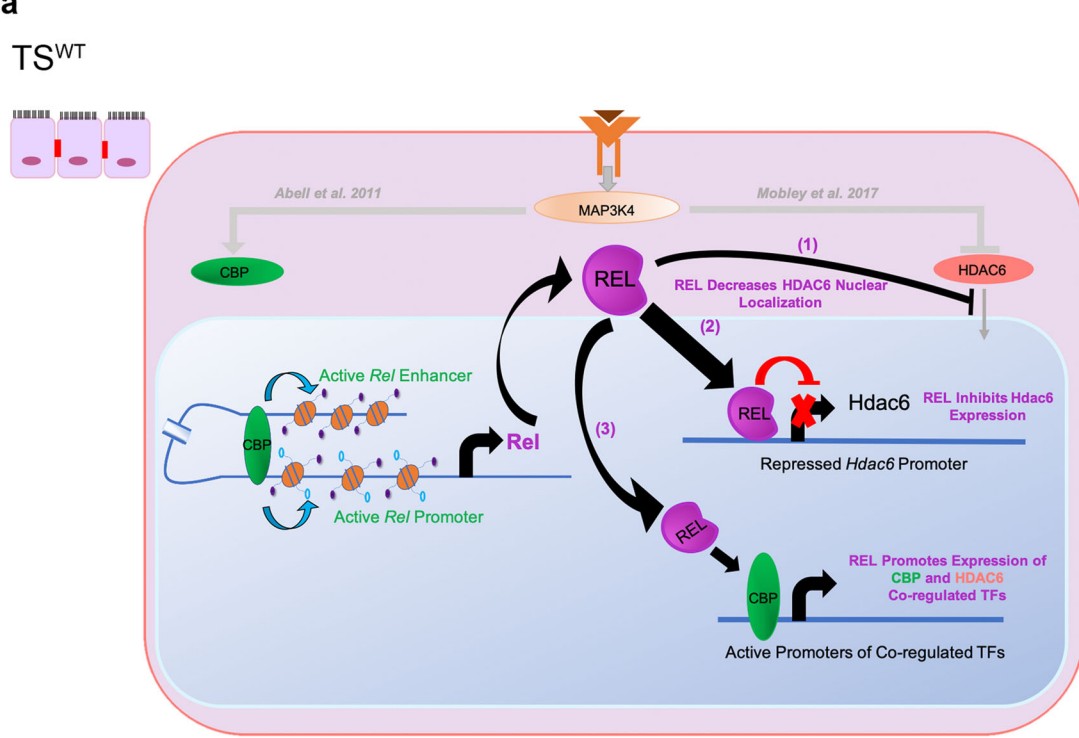

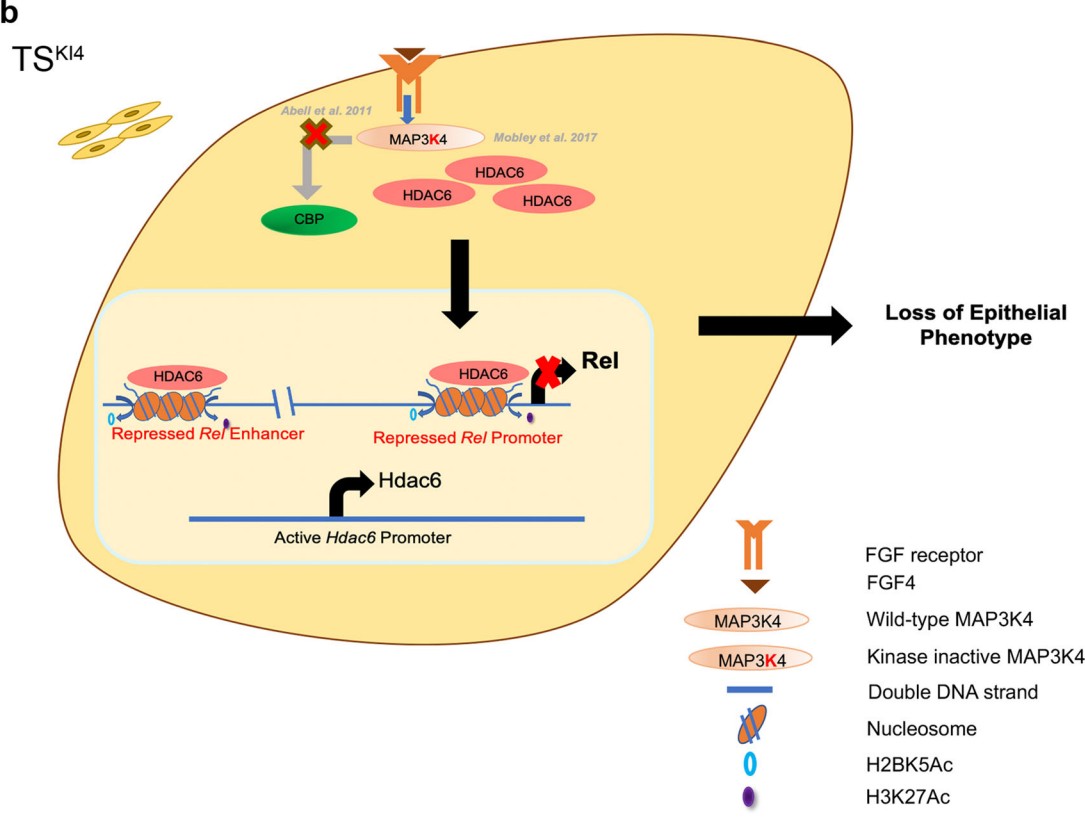

**Fig. 8 Developmental program coordinates switching between epithelial and mesenchymal phenotypes in TS cells. a** Rel transcript expression in epithelial TS^WT cells is promoted by CBP binding and acetylation of histones on *Rel* regulatory regions, maintaining the epithelial state. REL is active and translocates to the nucleus. REL maintains the epithelial phenotype through three different mechanisms. (1) REL inhibits HDAC6 nuclear localization, decreasing nuclear HDAC6 levels. (2) REL binds the *Hdac6* promoter and represses Hdac6 expression. (3) REL promotes the expression of CBP/HDAC6 co-regulated transcription factors. **b** Rel transcript expression is repressed in mesenchymal TS^KI4 cells by HDAC6 deacetylation of histones on *Rel* regulatory regions, resulting in loss of the epithelial phenotype.

REL promotes expression of CBP and HDAC6 co-regulated TFs. Our findings are supported by in silico predictions suggesting that Rel expression correlates with Cdh1 expression[40]. One mechanism by which REL induces MET is through repression of Hdac6 where REL binds the Hdac6 promoter in TS[WT] cells. Our findings suggest that REL may act as a physical repressor of Hdac6 expression, preventing HDAC6 induction of EMT. We predict REL induces MET through additional mechanisms, including direct regulation of CBP/HDAC6 co-regulated TFs. Co-regulation of Rel expression by CBP and HDAC6 coupled to REL repression of Hdac6 expression provides a coordinated biological program to control switching between different phenotypic states.

In summary, we define a program controlling switching between epithelial, intermediate, and mesenchymal states in TS cells. In this program, MAP3K4, CBP, and HDAC6 coordinate H2BK5Ac and gene expression of only 183 genes for selection of phenotypic states. REL is a key regulator of the phenotypic state whose expression is controlled by direct binding of CBP and HDAC6 to Rel regulatory regions. We also demonstrate the reciprocal regulation by REL of TFs co-regulated by CBP and HDAC6. Further, REL represses the chromatin modifier HDAC6 through direct binding to the Hdac6 promoter. Together, MAP3K4, CBP, HDAC6, and REL function as a tunable program to control the phenotypic state. A key question remaining relates to the potential for direct binding of REL to promoters or enhancers of co-regulated TFs. Future experiments will also focus on the mechanisms by which the other co-regulated TFs control cellular phenotype. Together, our work provides insight into the mechanisms controlling cellular decisions to establish different phenotypes. As small molecule drugs are currently available for CBP, HDAC6, and REL, coordination of their activities using these drugs may lead to strategies to control switching between cellular phenotypes.

## Methods

**Cell lines and culture conditions.** TS[WT] cells (male) and TS[KI4] cells (female) were isolated previously by Abell et al. (2009) from crosses of mice heterozygous for a targeted mutation in MAP3K4[5]. This substitution of the active site lysine at position 1361 with arginine inactivates MAP3K4 kinase activity[5,41]. TS cells were cultured in the absence of mouse embryonic fibroblast feeders in 30% TS cell media, which includes RPMI 1640, 20% heat inactivated fetal bovine serum (FBS), 1% penicillin and streptomycin (PS), 1% L-glutamine, 1% sodium pyruvate, and 100 μM β-mercaptoethanol and in 70% TS cell media conditioned by mitotically inactivated mouse embryonic fibroblasts[5]. To maintain stemness, TS cell media was supplemented with FGF4 (37.5 ng/ml) and heparin (1 μg/ml). To induce differentiation, TS cells were cultured in the absence of conditioned media, FGF4, and heparin[5]. The human cell lines HEK293T cells (fetal female), HMECs (adult female), and SUM159 cells (adult female) were a kind gift from Dr. Gary Johnson (UNC, Chapel Hill). HEK293T cells were cultured using Dulbecco's modified essential medium containing 10% FBS and 1% PS. HuMEC ready medium (Thermo Fisher Scientific) containing 5% FBS, 1% PS with HuMEC supplement and bovine pituitary extract were used to culture HMECs. Ham's F12 medium (Thermo Fisher Scientific) supplemented with 5% FBS, 1% PS with 5 μg/ml insulin and 1 μg/ml hydrocortisone was used to culture SUM159 cells[42]. All cells were cultured in a humidified atmosphere at 37C containing 5% CO2 (human) or 7% CO2 (mouse)[42]. All cell lines were examined and found negative for mycoplasma.

**Plasmids.** Using the Gateway Cloning system, FLAG-tagged human REL and RELB expressing lentiviral constructs were generated by cloning pDONR223- REL (clone# 53942 BC117191, Vidal human ORFeome (Version 5.1)) and pENTR223-RELB plasmid (clone# HsCD00515954, DNASU plasmid repository) respectively into a lentiviral FLAG-tagged destination vector[43]. A doxycycline (DOX) inducible lentiviral construct expressing human REL (Tet-on inducible REL) was also created using Gateway Cloning. The REL coding sequence was cloned from pDONR223-Rel into the doxycycline-inducible system pCW57.1 vector (Addgene). Control TS cells were created using lentiviral pLKO plasmid. Crebbp shRNA knockdown cells were created using lentiviral TRCN0000012723 and TRCN0000012727 plasmids (Dharmacon). The Firefly 3X NF-κB luciferase reporter gene was kindly provided by Dr. A.S. Baldwin (UNC Chapel Hill)[44].

**Lentiviral production and infection.** Lentivirus was produced in HEK293T cells as previously described[42]. Briefly, HEK293T cells were cultured until reaching 50–60% confluency. Using calcium phosphate, cells were co-transfected with packaging plasmid (psPAX2) and envelope plasmid (pMD2.G)(Addgene), and either pLKO, FLAG-tagged expression constructs, shRNAs, or Tet-on inducible constructs. Viral supernatants were harvested after 48 and 72 h of transfection using ultracentrifugation. Viral pellets were resuspended in 100 μl of TS media with growth factors. TS cells were infected with the lentiviruses as previously described[3]. Stably transduced cells were selected using 1 μg/ml puromycin. To express REL transiently, TS[KI4] cells transduced with doxycycline inducible REL lentivirus (TS[KI4R]) were treated with 1 μM 5-aza-2′-deoxycytidine (Aza) for 48 h and 2 μg/ml Dox for 96 h. Then, RNA and proteins were harvested for further analysis. To establish the stable expression of Dox inducible REL in TS[KI4R] cells, cells were treated with 1 μM Aza for 48 h and 2 μg/ml Dox for 96 h. Then, cells were cultured and treated continuously every other day in the presence of Dox alone.

**RNA sequencing.** Libraries were prepared by the High Throughput Genomic Sequencing Facility (HTSF) at the University of North Carolina at Chapel Hill according to the manufacturer's instructions (Illumina). The Ensembl mouse reference genome GRCm38 build 84 was used to align all reads generated by Illumina HiSeq[45]. The reference genome index was created using Bowtie2 and the RNA-seq reads for each sample were aligned to the reference genome using TopHat2[46,47]. The aligned reads for each sample were then independently processed using GFOLD and the reference gene annotation to obtain normalized reads per kilobase million (RPKM) values for each gene[48]. All three aforementioned packages Bowtie2, TopHat2, and GFOLD, were run with default parameters. Zero RPKM values were converted to "epsilon", a small number equaling 2.2204e-16 to avoid division-by-zero errors during fold-change calculations.

**RNA sequencing analyses using DAVID.** Functional enrichment analysis for the 183 MAP3K4, CBP, HDAC6, and H2BK5Ac dependent genes was performed using DAVID[49,50]. Only 92 of 183 genes were annotated. Enrichment of categories in the Molecular Function Gene Ontology (GO) terms were examined using a threshold EASE score of 0.1.

**Microarray data.** Total cellular RNA isolated using the RNeasy Plus Minikit (Qiagen) was labeled and hybridized to Affymetrix GeneChip Mouse Gene 2.0 ST Arrays at the Feinstone Center at the University of Memphis according to manufacturer's instructions (Affymetrix). Functional enrichment analysis was performed on genes whose expression is upregulated in TS[KI4] cells re-expressing REL (TS[KI4R]) relative to TS[KI4] cells using DAVID[49,50]. Genes with a (log2(fold change) ≥ 2.0 and with an FDR < 0.05 were examined. Enrichment of categories in the Cellular Component GO terms were examined using a threshold EASE score of 0.1.

**Cell lysis, immunoprecipitation, and Western blotting.** TS cell whole-cell, nuclear, cytoplasmic, and histone lysates were prepared as previously described[5,41]. Briefly, whole-cell lysates were generated after disrupting the cells using buffer A (20 mM Tris pH 7.4, 150 mM NaCl, 1 mM EDTA, 1 mM EGTA and 1% Triton) with protease inhibitors (1 mM PMSF and 100 KIU/ml aprotinin) and phosphatase inhibitors (2 mM sodium vanadate and 20 mM sodium fluoride)[3]. Nuclear proteins were extracted in RIPA buffer (buffer A, 0.1% sodium dodecyl sulfate, and 0.1% sodium deoxycholate) with protease and phosphatase inhibitors as described above. Cytoplasmic proteins were extracted in 0.5% Triton in 1X phosphate buffered saline (PBS) with protease and phosphatase inhibitors as described above[3]. Then, histones were extracted from the remaining nuclear pellets in 0.2 N HCl overnight with shaking at 4C. Western blots were imaged using Bio-Rad ChemiDoc version 2.3.0.07 and Image lab software version 5.2.1. Uncropped blots are presented in Supplementary Fig. 7. All antibodies used for Western blotting analysis are listed in Supplementary Table 2.

**Real-time quantitative PCR.** TS cell total cellular RNA was isolated using an RNeasy Plus Minikit (QIAGEN), and cDNA was prepared from 3 μg of total RNA using the High-Capacity cDNA reverse transcription kit (ThermoFisher Scientific) as previously described[42]. Gene expression was measured using iTaq Universal SYBR Green Supermix (Bio-Rad) and the Bio-Rad CFX96 Touch. Gene expression levels were calculated using the $2^{-\Delta\Delta CT}$ method and normalized to Actb, Rps11, or Gapdh as specified in the figure legends. Primers used are specified in Supplementary Table 3.

**NF-κB luciferase reporter assay.** TS cells were transfected using the Neon Transfection System (Life Technologies). Briefly, $1 \times 10^5$ cells were resuspended in 10 μl of R buffer containing 500 ng of the Firefly 3X NF-κB luciferase and 50 ng of Renilla luciferase constructs (Promega). Electroporation was carried out at 1400 V with 10 ms pulse width for 3 pulses. Cells were cultured in antibiotic free media. Twenty-four-hours post transfection, Firefly luciferase activity was determined by a dual luciferase assay system according to the manufacturer's instructions (Promega) and normalized to Renilla activity to account for differences in transfection efficiency. Luciferase activity was measured with a Synergy H1MD plate reader (BioTek).

**Invasion assays and isolation of invasive trophoblasts.** Matrigel invasion assays were carried out as previously described[3,42]. Briefly, TS cells were seeded on growth factor reduced Matrigel coated 8 μm pore transwell inserts. Invasion assays were terminated after 48 h. Non-invading cells were removed from the top of transwells by swabbing and washing with 1X PBS. $T^{INV}$ cell RNA was isolated from the bottom of tranwells by lysing cells directly in RNAeasy RLT lysis buffer (Qiagen). To measure cell invasiveness, $T^{INV}$ cells at the bottom of transwells were fixed in 3% paraformaldehyde for 10 min and nuclei were stained with DAPI (2 μg/ml) for 30 min. Using an EVOS epifluorescence microscope, five 10X fields were captured per transwell. The cell number in each image was counted manually. Graphs were plotted using Prism7 (version 7.0, GraphPad Prism software).

**Transwell solute flux assay.** Barrier function was measured as previously described[4]. Briefly, cells were seeded in a 24 well plate on 0.4 μm pore size PET membrane Transwells (Corning Life Sciences). After three days, cells were confluent and 70 kDa rhodamine B isothiocynate-Dextran dye (1 mg/ml) was added to the top of each transwell. Fluorescence of dye in the lower chamber was measured using a Synergy H1MD plate reader (BioTek).

**Immunofluorescence staining and confocal microscopy.** Immunofluorescence staining was performed as previously described[4,42]. Briefly, cells were plated on acid washed glass coverslips in six-well culture plates for two or three days. Cells were fixed with 3% paraformaldehyde in 1X PBS for 10 min, permeabilized using 0.1% Triton for 3 min, and blocked using 10% goat serum for 1 h at room temperature (RT). Cells were incubated with anti-E-cadherin primary antibody overnight at 4C. After washing five times with 1X PBS, coverslips were incubated with DAPI (0.1 μg/ml), Dy-Alexa 488 (1:500) (Thermo Fisher Scientific) for one hour at RT. Coverslips were washed for 30 min and mounted on slides using media containing 90% glycerol and 10% 1 M Tris pH 7.5. Images were captured using a Nikon A1 laser scanning confocal microscope and a 40X Plan Fluor 1.3 NA oil objective. Images were captured with lasers at 408 and 488 nm. Individual z-stacks were obtained. Two-dimensional images were displayed using Extended Depth of Focus (EDF- Nikon NIS Elements) at all wavelengths.

**Live cell imaging.** Live cell imaging was performed as previously described using a Lionheart FX automated live cell imager (BioTek)[42]. In brief, cells were cultured for 48 h in 6-well tissue culture dishes before imaging. Using BioTek Gen5 software, we defined the beacons for each well. Then, the 3 ×3 images were captured through the 20X objective every 5 min for 28 h. Images were stitched and movies were created using BioTek Gen5 software.

**In silico *Hdac6* promoter analysis for REL binding sites.** The *Hdac6* promoter region was predicted based on the position of CpG islands and H3K4me3 marks using the UCSC genome browser. The promoter region was defined as −1 bp to −1000 bp upstream of the transcription start site. Four potential REL binding sites were predicted on the *Hdac6* promoter region using MATCH™ tool (public version 1.0)[51]. These sites were found −61, −90, −194, and −436 bp upstream of the *Hdac6* transcription start site. Sites at −61, and −90 bp were predicted to be binding sites for all NF-κB members, whereas sites at −194, and −436 bp were predicted to be specific REL binding sites. Based on the close proximity of the sites located at −194 bp and −436 bp, we designed primers to detect this region using anti-REL ChIP-PCR.

Forward primer: CTT CAG GCG CAA ACC ACA C.
Reverse primer: TCC GGG TCT CGA ACT AGG G

**Chromatin immunoprecipitation coupled to PCR.** ChIP experiments in TS cells were performed using a Simple ChIP Plus Enzymatic Chromatin IP Kit (9005S, Cell Signaling) according to the manufacturer's instruction. Briefly, TS cells were crosslinked in 1% paraformaldehyde for 10 min, and the reaction was stopped by adding glycine. Cells were lysed in lysis buffer A, nuclei were pelleted, and treated with micrococcal nuclease (0.5 μl per $4 \times 10^6$ cells) for 20 min at 37C. Digestion was terminated using 0.05 M EDTA. Then, samples were sonicated for 20 sec at 40% and then 30 s at 60% to disrupt the nuclear membranes. The supernatant was transferred to new tubes after centrifugation. Samples were treated with RNase and proteinase K to digest RNA and proteins in the samples. DNA-protein crosslinks were reversed at 65C for 90 min. Then, DNA was purified using MinElute columns (Qiagen). DNA was analyzed using agarose gels and quantified using a Qubit 3.0 Fluorometer (Thermo Fisher Scientific). Immunoprecipitation was performed using 10 μg of DNA and 1 μg of antibodies overnight at 4C with rotation, followed by incubation with Protein G magnetic beads for 2 h at 4C. Beads were washed, and chromatin was eluted. DNA-protein crosslinks were reversed, and DNA fragments were purified. The purified DNA was analyzed using real-time qPCR analysis and ChIP primers listed in Supplementary Table 4. qPCR analysis was performed using SsoAdvanced Universal SYBR Green Supermix (Bio-Rad) and Bio-Rad CFX96 Touch. Enrichment was calculated using the $2^{-\Delta\Delta CT}$ method and normalized to input control. ChIP experiments in HMECs and SUM159s were performed as previously described[42]. Primers used for ChIP-PCR in HMECs and SUM159s are specified in Supplementary Table 5.

**Statistics and reproducibility.** Statistical analyses were performed using Prism7 (version 7.0, GraphPad software). Prism7 was used to perform a two-tailed unpaired Student's *t* test. qPCR data for supplementary data files were analyzed using two-tailed unpaired Student's *t* test using the Bio-Rad CFX Maestro software (version 4.0). Information about statistical methods for all experiments including statistical tests used, value of *n*, and what *n* represents is defined in each figure legend. A *$p$-value < 0.05, **$p$-value < 0.01, ***$p$-value < 0.001, and ****$p$-value < 0.0001 were all considered statistically significant. One-way ANOVA was used for the statistical analyses of microarray gene expression data. Additional information related to statistical analyses and reproducibility is included in Supplementary Data 3.

**Reporting summary.** Further information on research design is available in the Nature Research Reporting Summary linked to this article.

## Data availability

The sequencing and microarray data were deposited in the NCBI Gene Expression Omnibus (GEO). The accession number for all new sequencing data reported in this paper is GEO: GSE148496. The accession number for microarray data reported in this paper is GEO: GSE148250. The accession number for previously reported promoter capture Hi-C data deposited in ArrayExpress is E-MTAB-6585. The raw source data for the main figures are included in the Supplementary Data 3 provided in Supplementary Information. Any other data not included in the paper or supplementary materials is available from the authors upon reasonable request. The corresponding author, Dr. Amy Abell, will provide the requested data.

## Code availability

Custom code was not created by the authors. Commercially available software from Bio-Rad was used to image and analyze gels and qPCR data. Bio-Rad ChemiDoc Version 2.3.0.07 commercially available software from Bio-Rad was used to run the Bio-Rad ChemiDoc Touch Imager. Bio-Rad Image Lab 5.2.1 commercially available software from Bio-Rad was used to analyze images of gels and blots acquired using the Bio-Rad ChemiDoc Touch Imager. Bio-Rad CFX Maestro software 4.0.2325.2 commercially available software from Bio-Rad was used to analyze qPCR data acquired with a Bio-Rad CFX96 Touch qPCR machine.

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

## Acknowledgements

A.N.A. is supported by the National Institutes of Health grant (GM116903) and by the Memphis Research Consortium. We thank Dr. Thomas Sutter and the Feinstone Center at the University of Memphis for microarray data generation and analyses. We thank Dr. Omar Skalli and the Integrated Microscopy Center at the University of Memphis for use of the Nikon A1 laser scanning confocal microscope.

## Author Contributions

N.A.M.S., D.R., C.H.P., and A.S. performed experiments. N.A.M.S., D.R., C.H.P., and A.N.A. designed experiments. S.R. and R.H. performed bioinformatics analyses of RNA-seq and ChIP-seq data sets. M.R.B. analyzed promoter capture Hi-C data. N.A.M.S., S.R., M.R.B, R.H., and A.N.A. wrote and edited the manuscript.

## Competing interests

The authors declare no competing interests.
