## [Peer Review File · Communications Biology]

Reviewers' comments:

Reviewer #1 (Remarks to the Author):

Manuscript Title: Epigenetic regulation of Rel expression by MAP3K4, CBP, and HDAC6, controls phenotypic switching. (# COMMSBIO-20-0163-T)

In this manuscript, the author claimed their works evident epigenetic program controlling switching between epithelial and mesenchymal phenotypes. The works seem to be a consequent relevant study of previous work (Cell Stem Cell. 2011 May 6;8(5):525-37.). Epigenetic program controlling switching between epithelial and mesenchymal phenotypes was well known characters, but the regulatory mechanisms are defined yet. In this manuscript, the author proposed Rel expression by MAP3K4, CBP, and HDAC6 controls phenotypic switching was still an phenomena observation, such as previous other works (2011, 2017, and 2019), and less mechanistically information were provided. Actually, in MAP3K4-mutated cells, many downstream signals were changed and interference EMT transition. Through simply gene expression and knock down by shRNAi in MAP3K4-mutated cells, partially activity of EMT recovered by downstream targets (such as Rel over expression) was predictable, as previous works by the same group.

Further, member of NF-kappa B (especially c-Rel and p52) were well known involved in EMT regulated by epigenetics regulation. So, the current study did not provide more information about the issue of "Epigenetic program controlling switching between epithelial and mesenchymal phenotypes". Totally, the current manuscript just provided the concept : "Rel, one of downstream target of MAP3K4 interferes EMT in trophoblast cells."

In addition, there are several basic defects (internal controls and approaches) in whole story. For example, the level of tubulin expression was regulated by HDAC6 , and showed different in MAP3K4-mutant cells, but the authors used it as internal control in immunoblotting to show equal loading (Fig 1, 5 7). More, ERK were used as internal control in Fig. 6. All of them interference the all of quantitative result in the manuscript.

Reviewer #2 (Remarks to the Author):

The authors previously found that MAP3K4 plays an important role in epithelial-mesenchymal transitions (EMT) in trophoblast cells. Further, they discovered MAP3K4 regulates EMT through acetyltransferase CBP and deacetyltransferase HDAC6. This manuscript is an extension of their previous findings. They used bioinformatics analysis to identify one transcription factor from the NFkB family, Rel, which could be co-regulated by MAP3K4/CBP/HDAC6. They have shown that Rel expression regulates EMT. Moreover, they have also shown that HDAC6 represses Rel transcription and that Rel also represses HDAC6 expression and decreases HDAC6 nuclear localization. Overall, this manuscript provides convincing data to support their conclusions. It is a well-written manuscript. It is suitable to be published in Communications Biology.

However, there are a couple of suggestions. First, Rel has been identified as a key downstream target for the MAP3K4/CBP/HDAC6 program to regulate EMT in this manuscript. What are the phenotypes of Rel knockout mice? Second, the authors have shown that HDAC6 regulates Rel expression and that Rel regulates HDAC6 expression and decreases HDAC6 nuclear localization in a mouse cell line. I wonder this data is also true in humans. Human HDAC6 harbors a unique SE-14 motif, which serves as a cytoplasmic anchor (Bertos et al, 2004 JBC Vol 279 pp48246-48254). Perhaps only a small portion of human HDAC6 is located in the nucleus. Thus, human HDAC6 and mouse HDAC6 may differ in regulating gene transcription. The authors may discuss the above issues in the discussion.

Reviewer #1 (Remarks to the Author):

Manuscript Title: Epigenetic regulation of Rel expression by MAP3K4, CBP, and HDAC6, controls phenotypic switching. (# COMMSBIO-20-0163-T)

In this manuscript, the author claimed their works evident epigenetic program controlling switching between epithelial and mesenchymal phenotypes. The works seem to be a consequent relevant study of previous work (Cell Stem Cell. 2011 May 6;8(5):525-37.). Epigenetic program controlling switching between epithelial and mesenchymal phenotypes was well known characters, but the regulatory mechanisms are defined yet. In this manuscript, the author proposed Rel expression by MAP3K4, CBP, and HDAC6 controls phenotypic switching was still an phenomena observation, such as previous other works (2011, 2017, and 2019), and less mechanistically information were provided. Actually, in MAP3K4-mutated cells, many downstream signals were changed and interference EMT transition. Through simply gene expression and knock down by shRNAi in MAP3K4-mutated cells, partially activity of EMT recovered by downstream targets (such as Rel over expression) was predictable, as previous works by the same group.

Further, member of NF-kappa B (especially c-Rel and p52) were well known involved in EMT regulated by epigenetics regulation. So, the current study did not provide more information about the issue of “ Epigenetic program controlling switching between epithelial and mesenchymal phenotypes”. Totally, the current manuscript just provided the concept : “Rel, one of downstream target of MAP3K4 interferes EMT in trophoblast cells.”

In addition, there are several basic defects (internal controls and approaches) in whole story. For example, the level of tubulin expression was regulated by HDAC6 , and showed different in MAP3K4-mutant cells, but the authors used it as internal control in immunoblotting to show equal loading (Fig 1, 5 7). More, ERK were used as internal control in Fig. 6. All of them interference the all of quantitative result in the manuscript.

Reviewer #2 (Remarks to the Author):

The authors previously found that MAP3K4 plays an important role in epithelial-mesenchymal transitions (EMT) in trophoblast cells. Further, they discovered MAP3K4 regulates EMT through acetyltransferase CBP and deacetyltransferase HDAC6. This manuscript is an extension of their previous findings. They used bioinformatics analysis to identify one transcription factor from the NFkB family, Rel, which could be co-regulated by MAP3K4/CBP/HDAC6. They have shown that Rel expression regulates EMT. Moreover, they have also shown that HDAC6 represses Rel transcription and that Rel also represses HDAC6 expression and decreases HDAC6 nuclear localization. Overall, this manuscript provides convincing data to support their conclusions. It is a well-written manuscript. It is suitable to be published in Communications Biology.

However, there are a couple of suggestions. First, Rel has been identified as a key downstream target for the MAP3K4/CBP/HDAC6 program to regulate EMT in this manuscript. What are the phenotypes of Rel knockout mice? Second, the authors have shown that HDAC6 regulates Rel expression and that Rel regulates HDAC6 expression and decreases HDAC6 nuclear localization in a mouse cell line. I wonder this data is also true in humans. Human HDAC6 harbors a unique SE-14 motif, which serves as a cytoplasmic

anchor (Bertos et al, 2004 JBC Vol 279 pp48246-48254). Perhaps only a small portion of human HDAC6 is located in the nucleus. Thus, human HDAC6 and mouse HDAC6 may differ in regulating gene transcription. The authors may discuss the above issues in the discussion.

Response to Reviewers

Reviewer #1 (Remarks to the Author):

Manuscript Title: Epigenetic regulation of *Rel* expression by MAP3K4, CBP, and HDAC6, controls phenotypic switching. (# COMMSBIO-20-0163-T)

In this manuscript, the author claimed their works evident epigenetic program controlling switching between epithelial and mesenchymal phenotypes. The works seem to be a consequent relevant study of previous work (Cell Stem Cell. 2011 May 6;8(5):525-37.). Epigenetic program controlling switching between epithelial and mesenchymal phenotypes was well known characters, but the regulatory mechanisms are defined yet. In this manuscript, the author proposed *Rel* expression by MAP3K4, CBP, and HDAC6 controls phenotypic switching was still an phenomena observation, such as previous other works (2011, 2017, and 2019), and less mechanistically information were provided. Actually, in MAP3K4-mutated cells, many downstream signals were changed and interference EMT transition. Through simply gene expression and knock down by shRNAi in MAP3K4-mutated cells, partially activity of EMT recovered by downstream targets (such as *Rel* over expression) was predictable, as previous works by the same group.

We thank the reviewer for their comments and suggestions. The reviewer makes an important point regarding the lack of novelty for the role of epigenetic regulation of EMT. There are many reviews on the topic. In the original manuscript, we overused the term “epigenetic,” and we did not clearly emphasize the many, new, previously unknown findings in the manuscript. In the revised manuscript, we have made the following changes to clearly emphasize the novelty of our findings and the mechanisms by which REL promotes MET. In addition, we have added two new figures (**Fig. 8** and **Supplementary Fig. 5**) and one table (**Supplementary Table 1**) to strengthen the impact of our findings. We have also modified **Figs. 2, 5, 6 and 7**, and **Supplementary Fig. 6** to include additional data.

We have changed the title to emphasize our novel discovery that *Rel* transcript and protein expression levels are **coordinated** by MAP3K4 regulation of both CBP and HDAC6. “Coordinate regulation of *Rel* expression by MAP3K4, CBP, and HDAC6 controls phenotypic switching.” Our previous work has focused on genes that were either regulated by CBP or HDAC6. Here we show that *Rel* is one of only 183 genes that are co-regulated together by CBP and HDAC6 mediated histone acetylation. Considering that CBP and HDAC6 individually regulate thousands of genes, we think it is very surprising that CBP and HDAC6 co-regulate so few genes in TS cells.

We have modified the Abstract to clearly focus on the new, previously unknown discoveries in the manuscript. Specifically, we explain the following:

1. Even though CBP and HDAC6 each individually regulate many thousands of genes, CBP and HDAC6 only co-regulate 183 genes. “Surprisingly, only 183 genes are co-regulated by MAP3K4, CBP, and HDAC6 mediated histone acetylation.”

2. Our finding showing regulation of *Rel* expression at the transcript level is very unusual, because *Rel* is constitutively expressed in most cell types. Similar to the other members of the NF- κ B family, *Rel* is regulated almost exclusively at the post-transcriptional level. REL protein resides in the cytoplasm, being held inactive by the repressor I κ B. “The highest-ranking co-regulated gene is the NF- κ B family member *Rel*. Although the NF- κ B family is primarily regulated post-transcriptionally, both CBP and HDAC6 control *Rel* transcript levels by binding *Rel* regulatory regions and controlling histone acetylation.”
3. “Re-expression of REL in mesenchymal-like TS^{K14} cells induces a mesenchymal-epithelial transition, demonstrating previously unknown roles for REL.” It is important to point out that REL has not previously been shown to induce MET. We are the first to identify REL as an inducer of MET. In contrast, NF- κ B family members have been shown in many publications to induce EMT.
4. REL physically binds the *Hdac6* promoter, and represses *Hdac6* expression and HDAC6 nuclear localization. REL has not been previously shown to regulate HDAC6. This feedback regulation by which REL represses *Hdac6* provides a mechanism for REL induction of MET in TS cells. Further, it shows how REL works with MAP3K4 to suppress HDAC6 induction of EMT. “Importantly, REL forms a feedback loop, blocking *Hdac6* expression and HDAC6 nuclear localization.”

We have modified the Introduction to focus on the novel identification of a small number of genes that are coordinately regulated by both CBP and HDAC6 mediated H2BK5Ac. We have completely removed the term “epigenetic” from the Introduction and the Results sections. In the Introduction, we explain how NF- κ B family members including REL are expressed constitutively in most cell types. The proteins of this family of transcription factors are localized in the cytoplasm bound to a physical repressor. Activation of NF- κ B results in degradation of the repressor and translocation of the NF- κ B family members to the nucleus. This additional information serves to help readers understand the novelty of our findings that *Rel* is regulated at the transcript level by CBP and HDAC6. The following are changes to the Introduction:

“Herein, we performed bioinformatics analyses of new RNA-seq data and published anti-H2BK5Ac chromatin immunoprecipitation (ChIP)-seq data to identify genes co-regulated by MAP3K4, CBP, and HDAC6. Of the thousands of genes independently regulated by either CBP or HDAC6, only 183 genes were co-regulated by both CBP and HDAC6 mediated regulation of histone acetylation. Within these co-regulated genes, 12% were DNA binding proteins that could be drivers of the epithelial maintenance program. The highest-ranking DNA binding protein was the NF- κ B family member *Rel*. This family of TFs is constitutively expressed at transcript and protein levels in most cell types and is regulated post-transcriptionally by a protein repressor found in the cytoplasm⁶. In contrast, we show that *Rel* transcript levels are controlled by both CBP and HDAC6. In epithelial TS^{WT} cells, *Rel* expression is promoted by CBP binding to both *Rel* promoter and predicted enhancer regions. In mesenchymal-like TS^{K14} cells, HDAC6 is bound to *Rel* regulatory regions, deacetylating histones H2BK5 and H3K27. *Rel* transcript expression is reduced in mesenchymal-like TS^{K14} cells deficient in MAP3K4 kinase activity, and re-expression of *Rel* in TS^{K14} cells induces MET. Finally,

we show that *Rel* expression induces a switch to an epithelial phenotype by promoting the expression of 71% of the CBP/HDAC6 co-regulated DNA binding proteins. Further, REL directly binds the *Hdac6* promoter, represses *Hdac6* expression, and reduces nuclear HDAC6 localization. In summary, we define a developmental program that coordinately regulates the expression of *Rel* to drive switching between mesenchymal and epithelial states in TS cells.”

In addition to modifying the text of the Results section, we provide additional data in one new supplementary table and two new figures. These data are included to increase the impact of our findings and to clearly distinguish our new findings from our previous work. The new data and their impact are described below.

1. To emphasize that CBP and HDAC6 often regulate the expression of different genes, we provide a new **Supplementary Table 1**. This table shows RPKM values for representative genes that are either CBP or HDAC6 dependent, but not co-dependent. By showing a few of these non co-regulated genes of which there are thousands, it emphasizes the importance of the 183 genes that we identified as co-regulated by both CBP and HDAC6 mediated histone acetylation. “We have previously shown that HDAC6 in the nuclei of TS^{K14} cells binds and represses the promoters of genes encoding tight junction proteins like Claudin6⁴. However, the expression of these genes is CBP independent (Supplementary Table 1).”

2. We provide a new **Supplementary Fig. 5**. (The previous Supplementary Fig. 5 is now Supplementary Fig. 6). In the new Supplementary Fig. 5, we extend the impact of our findings in murine TS cells to human cells. We examined REL, RELB, and HDAC6 in human mammary epithelial cells (HMECs) and mesenchymal claudin-low SUM159 breast cancer cells. Similar to our findings in TS cells, we show decreased *REL* transcript and increased *RELB* transcript in SUM159 relative to HMECs. Further, we show elevated nuclear HDAC6 expression in SUM159 relative to HMECs. Using anti-HDAC6 ChIP-PCR, we show enrichment of HDAC6 on the *REL* promoter in SUM159 relative to HMECs, and the concomitant decrease of H2BK5Ac on the *REL* promoter using anti-H2BK5Ac ChIP-PCR. These data suggest both a conserved mechanism for controlling *Rel* transcripts and a shared role for REL in both murine TS cells and human HMECs. This new information is found in the Results section.

3. To clarify the novelty of our findings and the mechanisms used by REL to promote the epithelial state, we provide a new **Fig. 8**. This figure is a graphical depiction that clearly cites our previously published findings in grey and delineates our new findings in color. The figure illustrates our new finding that *Rel* transcript expression is co-regulated by CBP and HDAC6 control of histone acetylation on *Rel* regulatory regions. Further, the figure clearly shows our discovery that REL induces the epithelial state. In addition, it states the new mechanisms we have identified by which REL induces MET by binding the *Hdac6* promoter, repressing *Hdac6* expression, and preventing HDAC6 localization to the nucleus. Finally, Figure 8 shows how REL promotes the expression of the other CBP/HDAC6 co-regulated TFs that are predicted to promote the epithelial state in TS cells. The following information is found in the figure legend for Fig. 8:

“Fig. 8 Novel developmental program coordinates switching between epithelial and mesenchymal phenotypes in TS cells.

a *Rel* transcript expression in epithelial TS^{WT} cells is promoted by CBP binding and acetylation of histones on *Rel* regulatory regions, maintaining the epithelial state. REL is active and translocates to the nucleus. REL maintains the epithelial phenotype through three different mechanisms. (1) REL inhibits HDAC6 nuclear localization, decreasing nuclear HDAC6 levels. (2) REL binds the *Hdac6* promoter and represses *Hdac6* expression. (3) REL promotes the expression of CBP/HDAC6 co-regulated transcription factors.

b *Rel* transcript expression is repressed in mesenchymal TS^{KI4} cells by HDAC6 deacetylation of histones on *Rel* regulatory regions, resulting in loss of the epithelial phenotype.”

Further, member of NF-kappa B (especially c-Rel and p52) were well known involved in EMT regulated by epigenetics regulation. So, the current study did not provide more information about the issue of “ Epigenetic program controlling switching between epithelial and mesenchymal phenotypes”. Totally, the current manuscript just provided the concept : “Rel, one of downstream target of MAP3K4 interferes EMT in trophoblast cells.”

We agree with the reviewer that the NF-κB family has been shown to induce EMT through epigenetic regulation of the expression of genes related to EMT. However, our new manuscript focuses on the epigenetic control of NF-κB expression itself during EMT and MET. In most cell types, NF-κB members are primarily regulated at the level of activity by controlling the translocation of constitutively expressed NF-κB dimers between the cytoplasm and nucleus. Our data show that the expression of the NF-κB family member *Rel* is regulated epigenetically at the transcript level. Further, we show for the first time that REL is the dominant NF-κB member in undifferentiated trophoblast stem cells. Importantly, we discovered that *Rel* expression induces MET in trophoblast stem cells. Our findings contrast strikingly with numerous previous studies that have demonstrated that NF-κB members induce EMT during tumor progression. We show the ability of REL to induce MET in TS cells is selective for REL, as expression of a different NF-κB family member RELB in TS cells induces EMT.

We define for the first time mechanisms by which REL induces MET. We show a feedback mechanism where REL binds directly to the *Hdac6* promoter regulating *Hdac6* transcript levels. In addition, we show that REL also controls HDAC6 protein localization, decreasing nuclear HDAC6 levels. These findings represent the first demonstration of REL regulating *Hdac6* expression and localization. It is important to point out that transient, doxycycline-induction of *REL* expression quickly repressed *Hdac6* expression and decreased HDAC6 nuclear localization. We have included our studies performed after transient doxycycline treatment in two new panels in **Supplementary Fig. 6**. The direct binding of REL to the *Hdac6* promoter combined with rapid, REL-induced changes in *Hdac6* expression and HDAC6 localization suggests REL directly regulates HDAC6 to drive MET. This feedback is an elegant mechanism to

modulate changes in cellular phenotype. The following is new text from the Discussion that discusses the mechanisms by which REL promotes the epithelial state:

“One mechanism by which REL induces MET is through repression of *Hdac6* where REL binds the *Hdac6* promoter in TS^{WT} cells. Our findings suggest that REL may act as a physical repressor of *Hdac6* expression, preventing HDAC6 induction of EMT. We predict REL induces MET through additional mechanisms, including direct regulation of CBP/HDAC6 co-regulated TFs. Co-regulation of *Rel* expression by CBP and HDAC6 coupled to REL repression of *Hdac6* expression provides a coordinated biological program to control switching between different phenotypic states.”

In addition, there are several basic defects (internal controls and approaches) in whole story. For example, the level of tubulin expression was regulated by HDAC6, and showed different in MAP3K4-mutant cells, but the authors used it as internal control in immunoblotting to show equal loading (Fig 1, 5 7). More, ERK were used as internal control in Fig. 6. All of them interference the all of quantitative result in the manuscript.

Specific loading controls for the Western blots were originally chosen based on several conditions, including the size of the proteins to allow ideally the use of the same membrane to examine both proteins. In addition, protein distribution in the cytoplasm versus nucleus was used as criteria for selection. As shown in our previous studies, total tubulin levels are not changed between our different cellular conditions. Although HDAC6 regulates the acetylation of tubulin, total tubulin levels are not altered. In the original manuscript submission, we used antibodies to total tubulin. To increase the rigor of the internal controls for our studies, we have performed additional Western blotting with the same lysates for multiple other proteins whose expression does not change in our system. Additional loading controls for each figure included in the revised manuscript are as follows:

Fig. 1: none, there are no Western blots in this figure.

Fig. 2: We now have Actin, total Tubulin, and LaminB1 as loading controls.

Fig. 5: We now have Actin, total Tubulin, and ERK as loading controls.

Fig. 6: We now have Actin, total Tubulin, and ERK as loading controls.

Fig. 7h, the frozen lysates were degraded, so we were unable to perform additional Western blots.

Fig. 7i: We now have both Actin and ERK as loading controls.

Reviewer #2 (Remarks to the Author):

The authors previously found that MAP3K4 plays an important role in epithelial-mesenchymal transitions (EMT) in trophoblast cells. Further, they discovered MAP3K4 regulates EMT through acetyltransferase CBP and deacetyltransferase HDAC6. This manuscript is an extension of their previous findings. They used bioinformatics analysis to identify one transcription factor from the NFkB family, Rel, which could be co-regulated by MAP3K4/CBP/HDAC6. They have shown that Rel expression regulates

EMT. Moreover, they have also shown that HDAC6 represses Rel transcription and that Rel also represses HDAC6 expression and decreases HDAC6 nuclear localization. Overall, this manuscript provides convincing data to support their conclusions. It is a well-written manuscript. It is suitable to be published in Communications Biology. However, there are a couple of suggestions.

We thank the reviewer for their comments and suggestions.

First, Rel has been identified as a key downstream target for the MAP3K4/CBP/HDAC6 program to regulate EMT in this manuscript. What are the phenotypes of Rel knockout mice?

Deletion of *Rel* in mice does not cause embryonic lethality. However, co-deletion of both *Rel* and *RelA* in mice results in earlier embryonic lethality than deletion of *RelA* alone. These studies suggest compensation by family members, hindering the identification of specific roles of each family member during development. Adult mice with the deletion of *Rel* show defects in lymphoid function and liver regeneration. *Rel* knockout mice have impaired expansion and survival of CD8⁺ T cells, suggesting a role for REL in lymphoid function. In addition, *Rel* knockout mice have an impaired ability to regenerate the liver after injury. Similar to others, we predict that the number of functional roles of REL have been underestimated due to compensatory mechanisms by other family members. The following is found in the revised Discussion:

“Although single deletion of *RelA*, *Ikkβ*, or *NEMO* results in embryonic lethality, deletion of two family members leads to more deleterious phenotypes with earlier onset. For example, the single deletion of *RelA* in mice results in embryonic lethality at E15 due to liver degeneration^{29,30}. Deletion of *Rel* does not result in embryonic lethality^{31,32}. However, co-deletion of *RelA* and *Rel* results in earlier lethality at E13.5, suggesting *Rel* may compensate for *RelA*²⁵.”

Second, the authors have shown that HDAC6 regulates Rel expression and that Rel regulates HDAC6 expression and decreases HDAC6 nuclear localization in a mouse cell line. I wonder this data is also true in humans. Human HDAC6 harbors a unique SE-14 motif, which serves as a cytoplasmic anchor (Bertos et al, 2004 JBC Vol 279 pp48246-48254). Perhaps only a small portion of human HDAC6 is located in the nucleus. Thus, human HDAC6 and mouse HDAC6 may differ in regulating gene transcription. The authors may discuss the above issues in the discussion

We have previously published that HDAC6 is found in the nucleus in human claudin-low breast cancer cells (Raghu, 2019). Based on the Reviewer's suggestion, we wondered if HDAC6 regulates *REL* expression in human cells. To examine this question, we performed experiments comparing human mammary epithelial cells (HMECs) to human claudin low SUM159 breast cancer cells. HMECs are epithelial and SUM159s display a mesenchymal morphology. Nuclear HDAC6 expression was modestly higher in the SUM159s relative to the HMECs. Further, we found that REL expression was decreased and RELB expression increased in SUM159s relative to the HMECs. Using

ChIP-PCR, we measured enrichment of HDAC6 on the *REL* promoter with a concomitant decrease in H2BK5Ac enrichment in SUM159s relative to the HMECs. These data indicate that HDAC6 may regulate *REL* expression in human cells, suggesting a conserved mechanism for regulation of *REL* expression. These new data are in a new **Supplementary Fig. 5** and are described in the Results section as follows:

“As human HDAC6 contains a tetradecapeptide repeat domain not present in mouse HDAC6 that promotes cytoplasmic retention of human HDAC6, we wondered if HDAC6 regulates *REL* expression in human cells¹². Comparison of human mammary epithelial cells (HMECs) to mesenchymal claudin-low SUM159 breast cancer cells revealed decreases in *REL* transcript and increases in *RELB* transcript in SUM159s relative to HMECs (Supplementary Fig. 5a-c). Nuclear HDAC6 and RELB protein levels were also increased in SUM159s (Supplementary Fig. 5d). ChIP-PCR showed enrichment of HDAC6 on the *REL* promoter and a concomitant decrease in H2BK5Ac on the *REL* promoter in SUM159s (Supplementary Fig. 5e, f). These data suggest that HDAC6 represses *REL* expression in both mouse and human cells.”

In the revised manuscript, we have made the following additional changes. We have added one table and one figure to increase the impact of our findings. To emphasize the significance of the 183 co-regulated genes, we provide a new **Supplementary Table 1** that shows how CBP and HDAC6 can individually regulate the expression of different genes. This table shows RPKM values for representative genes that are either CBP or HDAC6 dependent, but not co-dependent. By showing a few of these non co-regulated genes of which there are thousands, it emphasizes the importance of the 183 genes that we identified as co-regulated by both CBP and HDAC6 mediated histone acetylation. We also provide a new **Fig. 8**. This figure is a graphical depiction that clearly cites our previously published findings in grey and delineates our new findings in color. The figure illustrates our new finding that *Rel* transcript expression is co-regulated by CBP and HDAC6 control of histone acetylation on *Rel* regulatory regions. Further, the figure clearly shows our discovery that REL induces the epithelial state. In addition, it states the new mechanisms we have identified by which REL induces MET by binding the *Hdac6* promoter, repressing *Hdac6* expression, and preventing HDAC6 localization to the nucleus. Finally, Fig. 8 shows how REL promotes the expression of the other CBP/HDAC6 co-regulated TFs that are predicted to promote the epithelial state in TS cells.

a

Fold Change	TS ^{KI4R} /TS ^{KI4}	p-value
Crebl2	-1.0	0.700675
Ets1	+1.5	0.003677
Gli1	+1.0	0.597593
Hivep2	+2.6	0.000504
Hoxa1	+1.0	0.923662
Id2	+2.9	0.000024
Lhx6	+1.1	0.168735
Pou6f1	+1.2	0.105356
Rfx3	+1.1	0.199638
Runx1	+1.1	0.140903
Tlx1	-1.1	0.212118
Zfp672	+1.1	0.038137
Zfp810	-1.0	0.876777
2610008E11Rik	-1.2	0.168771

b

c

d

e

f

g

h

i

j

Supplementary Table 1: Representative genes whose expression is either **CBP** dependent or **HDAC6** dependent

RPKM Gene name	TS ^{WT}	TS ^{WTCBPsh}	TS ^{KI4}	TS ^{KI4H6sh}
Atf4	146.866	109.348	89.196	92.168
Irx3	56.216	33.050	39.216	24.125
Jdp2	21.460	7.531	2.764	2.630
Itgb2	0.248	0.016	0.022	0.018
Cldn6	122.076	152.681	27.753	139.987
Col4a2	61.2856	66.955	15.413	39.712
Tjp2	46.650	51.626	34.502	57.735
Ocln	0.391	0.930	0.189	1.039

REVIEWERS' COMMENTS:

Reviewer #1 (Remarks to the Author):

No more concern about the manuscript.

Reviewer #2 (Remarks to the Author):

The authors have addressed my concerns very well. Thank you.